# Variational Inference for Discriminative Learning with Generative Modeling of Feature Incompletion

**Kohei Miyaguchi, Takayuki Katsuki, Akira Koseki & Toshiya Iwamori**
IBM Research – Tokyo
miyaguchi@ibm.com, {kats,akoseki,iwamori}@jp.ibm.com

## Abstract

We are concerned with the problem of distributional prediction with incomplete features: The goal is to estimate the distribution of target variables given feature vectors with some of the elements missing. A typical approach to this problem is to perform missing-value imputation and regression, simultaneously or sequentially, which we call the generative approach. Another approach is to perform regression after appropriately encoding missing values into the feature, which we call the discriminative approach. In comparison, the generative approach is more robust to the feature corruption while the discriminative approach is more favorable to maximize the performance of prediction. In this study, we propose a hybrid method to take the best of both worlds. Our method utilizes the black-box variational inference framework so that it can be applied to a wide variety of modern machine learning models, including the variational autoencoders. We also confirmed the effectiveness of the proposed method empirically.

## 1 Introduction

We address the problem of prediction with incomplete data, which often arises in real-life data due to the lack of data collecting resources and/or privacy concerns. For example, consider analyzing electronic health records. The input variables of the prediction are patients' demographic characteristics and history of medical measurement data obtained with various instruments and the target variable is the survival time of patients. Thus, some of these features are not available depending on patients due to, e.g., non-standardized medical equipment, legal regulation and privacy concerns. More formally, each input is an incomplete set of features represented with a tuple $\mathbf{u} = (\tilde{\mathbf{x}}, \mathbf{m})$ such that $\tilde{\mathbf{x}} \in \mathbb{R}^{d_x}$ is the corrupted version of the complete input $\mathbf{x} \in \mathbb{R}^{d_x}$ and $\mathbf{m} \in \{0, 1\}^{d_x}$ indicates the missing entries of $\tilde{\mathbf{x}}$: For all $j \in [d_x]$, $\mathbf{m}_j = 1$ if and only if the $j$-th feature is missing. In particular, missing entries are filled with zero and non-missing entries are identical to the one in the complete input, $\tilde{\mathbf{x}}_j = (1 - \mathbf{m}_j)\mathbf{x}_j$. Given the incomplete features $\mathbf{u}$, we want to predict the outcome $\mathbf{y} \in \mathbb{R}^{d_y}$, which is modeled with a predictive distribution $\mathbf{y} \sim p(\mathbf{y}|\mathbf{u}, \theta)$. Here, the parameter $\theta$ is unknown and should be learned from data.

One straightforward way to deal with such incompletion is to incorporate generative models: The missing values in $\tilde{\mathbf{x}}$ are imputed with some generative model and the target $\mathbf{y}$ is predicted based on the result of the imputation. Sometimes the imputation is done implicitly or simultaneously with the prediction. An advantage of this method lies in that the meaning of $\mathbf{m}$ is inherently incorporated. Without such context, algorithms must learn by themselves that $\mathbf{m}$ is actually indicating missingness (if that is relevant to the prediction task), which incurs extra generalization errors. In other words, appropriate generative models make the prediction robust to the feature incompletion.

On the other hand, the discriminative models that directly learn the conditional density $p_\theta(\mathbf{y}|\mathbf{u})$ sometimes exhibit superior performance (Kuhn et al., 2013) due to their alignment in terms of the objective. This is especially the case when the feature incompletion is not a dominant factor (Lasserre et al., 2006). However, it is not as straightforward as generative models to incorporate the context of missingness into discriminative models and therefore their prediction may be less robust to the feature incompletion.

To best of our knowledge, there have been no general solution to this dilemma of the generative and the discriminative approaches, i.e., generic method of *discriminative inference with incomplete features for generative models*, which we abbreviate as *DIG*. In particular, although there have been similar efforts made in the literature, the previous methods are limited in terms of the applicable models. For example, several DIG methods have been proposed for exponential families (Ghahramani & Jordan, 1994; Smola et al., 2005) and Gaussian processes (Pacheco et al., 2014), but they cannot handle modern machine-learning architectures such as deep neural networks.

This motivates us to seek for general and widely applicable frameworks that solve the dilemma. To this end, we propose a new variational approximation and a new variance reduction technique, which enable us to perform DIG within the black-box variational inference (VI) framework (Jordan et al., 1999; Zhang et al., 2018). We have also empirically confirmed the effectiveness of the proposed method through numerical experiments, employing the variational autoencoders (VAEs) (Kingma & Welling, 2013) as the base generative model.

The rest of the paper is organized as follows. In Section 2, we introduce the mathematical notation and the technical challenges in DIG with flexible models. Then, in Section 3, we present our method adopting VI in the DIG setting. In Section 4, we show the experimental results that confirm the effectiveness of the proposed method. In Section 5, we review and discuss the related work. Finally, in Section 6, we conclude the paper with some remarks and future perspectives. All the proofs for theoretical statements are given in the appendix.

## 2 PRELIMINARY

In this section, we introduce the mathematical formulation of the DIG method and then review some background of its technical challenges.

### 2.1 DIG FORMULATION

Let $i \in [n]$ denote the index of instances, where $n$ is the number of data and $[n]$ denotes the set of integers $\{1, 2, \ldots, n\}$. Let $\mathbf{y}^{[n]} := \{\mathbf{y}^i \in \mathbb{R}^{d_y}\}_{i \in [n]}$ and $\mathbf{x}^{[n]} := \{\mathbf{x}^i \in \mathbb{R}^{d_x}\}_{i \in [n]}$, be the sets of the target and the complete feature vectors, where $d_y$ and $d_x$ are the dimensionalities of vectors. We assume we can only observe the set of incomplete feature vectors $\mathbf{u}^{[n]} := \{\mathbf{u}^i := (\tilde{\mathbf{x}}^i, \mathbf{m}^i)\}_{i \in [n]}$ instead of $\mathbf{x}^{[n]}$, where $\tilde{\mathbf{x}}^i \in \mathbb{R}^{d_x}$ is the corrupted version of $\mathbf{x}^i$ and $\mathbf{m}^i \in \{0, 1\}^{d_x}$ is the indicator of corruption ('mask vector') for all $i \in [n]$. Each mask vector $\mathbf{m}^i$ indicates which entry of $\tilde{\mathbf{x}}^i$ is corrupted; We have $\tilde{\mathbf{x}}^i_j = (1 - \mathbf{m}^i_j)\mathbf{x}^i_j$ for all $i \in [n]$ and all $j \in [d_x]$.

**Generative model.** Suppose that the generative process of the observables $(\mathbf{y}^{[n]}, \mathbf{u}^{[n]})$ is modeled with a parametrized family of instance-wise identical distributions. That is, the joint probability density of $(\mathbf{y}^{[n]}, \mathbf{u}^{[n]})$ is factored as

$$p(\mathbf{y}^{[n]}, \mathbf{u}^{[n]}|\theta) = \prod_{i \in [n]} p(\mathbf{y}^i, \mathbf{u}^i|\theta), \tag{1}$$

where the function $p(\mathbf{y}^i, \mathbf{u}^i|\theta)$ on the RHS denotes the common density function across instances $i \in [n]$ indexed with any parametric models $\theta$ (e.g., by neural networks).

**Discriminative objective.** The goal of DIG is to maximize the conditional likelihood of the corrupted data,

$$\mathcal{L}^{[n]}_{\mathbf{y}|\mathbf{u}}(\theta) := \ln p(\mathbf{y}^{[n]}|\mathbf{u}^{[n]}, \theta) = \sum_{i \in [n]} \ln p(\mathbf{y}^i|\mathbf{u}^i, \theta). \tag{2}$$

Now, let $\mathcal{L}^i_{\mathbf{y}|\mathbf{u}}(\theta) := \ln p(\mathbf{y}^i|\mathbf{u}^i, \theta)$ denote the $i$-th summand. Following Bayes' rule, each $\mathcal{L}^i_{\mathbf{y}|\mathbf{u}}(\theta)$ is computed with a difference of log likelihoods

$$\mathcal{L}^i_{\mathbf{y}|\mathbf{u}}(\theta) = \mathcal{L}^i_{\mathbf{y},\mathbf{u}}(\theta) - \mathcal{L}^i_{\mathbf{u}}(\theta), \tag{3}$$

where $\mathcal{L}^i_{\mathbf{y},\mathbf{u}}(\theta) := \ln p(\mathbf{y}^i, \mathbf{u}^i|\theta)$ and $\mathcal{L}^i_{\mathbf{u}}(\theta) := \ln \int p(\mathbf{y}^i, \mathbf{u}^i|\theta) \, d\mathbf{y}^i$. Since we employ gradient-based optimization methods, it suffices to consider (approximate) evaluation of the value and the gradient

of the instance-wise objectives $\mathcal{L}^i_{\mathbf{y}|\mathbf{u}}(\theta)$ separately. Thus, in the following, we limit our focus to the separate objective and omit the index $i$ if there is no risk of ambiguity.

## 2.2 DIG with Latent Variable Models (DIG-LVM)

We give further details of DIG with a specific class of generative models, namely the latent variable models (LVMs). LVM is one of the most common and expressive generative models. In the context of DIG, the generative density function is given by

$$p(\mathbf{y}, \mathbf{u}|\theta) = \int p(\mathbf{y}, \mathbf{u}, \mathbf{z}|\theta) \, \mathrm{d}\mathbf{z}, \tag{4}$$

where $\mathbf{z}$ is the instance-wise latent variable, which is expected to capture some higher-order information of the observables. For example, the variational autoencoders (VAEs) (Kingma & Welling, 2013), the generative adversarial networks (GANs) (Goodfellow et al., 2014) and the normalizing flow models (Rezende & Mohamed, 2015) are instances of LVM.

**Modeling feature-corruption process with LVM.** One advantage of LVM is that complex generative processes can be modeled with relatively simple joint density $p(\mathbf{y}, \mathbf{u}, \mathbf{z}|\theta)$. This is especially beneficial in modeling missing data: The full joint density $p(\mathbf{y}, \mathbf{u}, \mathbf{z}|\theta)$ can be easily designed to reflect one's belief on the corruption process such as MCAR and MNAR (Rubin, 1976) through the factorization of the density (e.g., see Collier et al. (2020)), while maintaining the expressibility of LVM. For example, the MCAR process can be modeled with

$$p(\mathbf{y}, \mathbf{u}, \mathbf{z}|\theta) = p(\mathbf{m}) \cdot p(\mathbf{z}) \cdot p(\mathbf{y}|\mathbf{z}, \theta) \cdot \prod_{j=1}^{d_x} \{p(\tilde{\mathbf{x}}_j|\mathbf{z}, \theta)\}^{1-\mathbf{m}_j}, \tag{5}$$

where $p(\mathbf{m})$ is the true marginal density function of $\mathbf{m}$, which is unknown but cancels out in the DIG objective (Equation 3), and $p(\mathbf{z})$ is an arbitrary fixed prior of the latent variable (such as Gaussian). On the other hand, the MNAR process can be modeled with

$$p(\mathbf{y}, \mathbf{u}, \mathbf{z}|\theta) = p(\mathbf{m}) \cdot p(\mathbf{z}) \cdot p(\mathbf{y}|\mathbf{z}, \mathbf{m}, \theta) \cdot \prod_{j=1}^{d_x} \{p(\tilde{\mathbf{x}}_j|\mathbf{z}, \mathbf{m}, \theta)\}^{1-\mathbf{m}_j}. \tag{6}$$

Note that these are just examples and it is possible to further incorporate domain knowledge on the generative process. An example of such models is the MAR process (Rubin, 1976) and its formulation is relegated to the appendix (Section G).

Under these factorizations, the log likelihoods on the RHS of Equation 3 for LVMs are computed as

$$\mathcal{L}_{\mathbf{y}, \mathbf{u}}(\theta) = \ln \int p(\mathbf{y}, \mathbf{u}, \mathbf{z}|\theta) \, \mathrm{d}\mathbf{z}, \qquad\qquad \mathcal{L}_{\mathbf{u}}(\theta) = \ln \int p(\mathbf{u}, \mathbf{z}|\theta) \, \mathrm{d}\mathbf{z}, \tag{7}$$

where both density functions in the integrals can be analytically computed under either of Equation 5 and Equation 6.

**Challenge: Inference with DIG-LVM.** To maximize Equation 3 with LVM, we have to evaluate the values and gradients of the likelihoods in Equation 7. If there are only *positive* log integrals in the objective, this can be approximately done with the variational inference (VI) framework (Jordan et al., 1999; Zhang et al., 2018) in favor of flexibility and scalability. However, with DIG, we also have a *negative* log integral, i.e., $-\mathcal{L}_{\mathbf{u}}(\theta)$. The optimization of such objective is relatively difficult since there have been no equivalent of VI in terms of the flexibility and scalability (see Section 5).

## 3 Variational Inference for DIG-LVM

Now we present the main result, the method of VI for DIG-LVM. In Section 3.1, we derive a variational approximation to the conditional likelihood given by Equation 3. Then, in Section 3.2 and 3.3, we derive a method for optimizing the variational approximation and its variant for variance reduction, respectively. Finally, in Section 3.4, we show the prediction procedure based on the optimized parameters.

### 3.1 VARIATIONAL LOWER/UPPER BOUNDS

In this section, we discuss an approximation of $\mathcal{L}_{\mathbf{y},\mathbf{u}}(\theta)$ and $\mathcal{L}_{\mathbf{u}}(\theta)$ in Equation 7 individually. Note that both are log integrals of the specific form

$$\mathcal{L}_{\mathbf{v}}(\theta) := \ln \int p(\mathbf{v}, \mathbf{z}|\theta) \, d\mathbf{z}, \tag{8}$$

where $\mathbf{v}$ represents either $(\mathbf{y}, \mathbf{u})$ or $\mathbf{u}$. Since such integrals constitute the objective $\mathcal{L}(\theta)$ with positive and negative signs, our goal is to derive both upper and lower bounds on Equation 8.

**Evidence Lower Bound (ELBO).** The lower bound is given in the standard way (Jordan et al., 1999; Zhang et al., 2018),

$$\mathcal{L}_{\mathbf{v}}(\theta) \geq \mathcal{L}_{\mathbf{v}}(\theta) - D_{\mathrm{KL}(\mathbf{z}|\mathbf{v})}(\phi\|\theta)$$
$$= \mathbb{E}_{\mathbf{z}\sim q(\mathbf{z}|\mathbf{v},\phi)} \left[ \ln \frac{p(\mathbf{v}, \mathbf{z}|\theta)}{q(\mathbf{z}|\mathbf{v}, \phi)} \right] =: \mathcal{L}_{\mathrm{ELBO}(\mathbf{v})}(\theta, \phi), \tag{9}$$

where $\phi$ is a variational parameter of a probability density function $q(\mathbf{z}|\mathbf{v}, \phi)$ and $D_{\mathrm{KL}(\mathbf{z}|\mathbf{v})}(\phi\|\theta) := \mathbb{E}_{\mathbf{z}\sim q(\mathbf{z}|\mathbf{v},\phi)}[\ln \frac{q(\mathbf{z}|\mathbf{v},\phi)}{p(\mathbf{z}|\mathbf{v},\theta)}]$ is the KL divergence of the parameters $\phi$ and $\theta$, We call $\mathcal{L}_{\mathrm{ELBO}(\mathbf{v})}(\theta, \phi)$ as *the evidence lower bound (ELBO)*. Since ELBO is an expectation of a tractable function, we approximate it with Monte-Carlo sampling.

**Evidence Upper Bound (EUBO).** To derive an upper bound, we start with applying the $\chi$-evidence upper bound (CUBO) (Dieng et al., 2017). For any real numbers $\alpha > 1$, CUBO is derived as follows:

$$\mathcal{L}_{\mathbf{v}}(\theta) \leq \mathcal{L}_{\mathbf{v}}(\theta) + (1 - \alpha^{-1}) D_{\alpha(\mathbf{z}|\mathbf{v})}(\theta\|\psi)$$
$$= \frac{1}{\alpha} \ln \mathbb{E}_{\mathbf{z}\sim q(\mathbf{z}|\mathbf{v},\psi)} \left[ \frac{p(\mathbf{v}, \mathbf{z}|\theta)}{q(\mathbf{z}|\mathbf{v}, \psi)} \right]^{\alpha} =: \mathcal{L}_{\mathrm{CUBO}(\mathbf{v})}(\theta, \psi), \tag{10}$$

where $\psi$ is a variational parameter of a probability density function $q(\mathbf{z}|\mathbf{v}, \psi)$ and $D_{\alpha(\mathbf{z}|\mathbf{v})}(\theta\|\psi) := \frac{1}{\alpha-1} \ln \int d\mathbf{z} \, p^{\alpha}(\mathbf{z}|\mathbf{v}, \theta) q^{1-\alpha}(\mathbf{z}|\mathbf{v}, \psi)$ denotes the $\alpha$-Rényi divergence of the parameters $\theta$ and $\psi$. Note here, unlike ELBO, CUBO is not unbiasedly approximated because of the logarithm wrapping the expectation. To address this issue, we apply another variational approximation with a divergence function $\Psi_{\alpha}(t) := (e^{\alpha t} - \alpha t - 1)/\alpha, t \in \mathbb{R}$: Since $\Psi_{\alpha}(t) \geq 0$ for all $t \in \mathbb{R}$, we have

$$\mathcal{L}_{\mathrm{CUBO}(\mathbf{v})}(\theta, \psi) \leq \mathcal{L}_{\mathrm{CUBO}(\mathbf{v})}(\theta, \psi) + \Psi_{\alpha}(\mathcal{L}_{\mathrm{CUBO}(\mathbf{v})}(\theta, \psi) - f(\mathbf{v};\xi))$$
$$= \frac{e^{-\alpha f(\mathbf{v};\xi)}}{\alpha} \mathbb{E}_{\mathbf{z}\sim q(\cdot|\mathbf{v},\psi)} \left[ \frac{p(\mathbf{v}, \mathbf{z}|\theta)}{q(\mathbf{z}|\mathbf{v}, \psi)} \right]^{\alpha} + f(\mathbf{v};\xi) - \frac{1}{\alpha} =: \mathcal{L}_{\mathrm{EUBO}(\mathbf{v})}(\theta, \psi, \xi), \tag{11}$$

where $\xi$ is a variational parameter of a real-valued function $f(\mathbf{v};\xi)$. We call the right-hand side as *the evidence upper bound (EUBO)*. Note that the expectation in EUBO is linearized and thus can be unbiasedly approximated with Monte-Carlo estimation.

**Conditional Evidence Lower Bound (CELBO).** Applying ELBO on $\mathbf{v} = (\mathbf{y}, \mathbf{u})$ and EUBO on $\mathbf{v} = \mathbf{u}$, we have a conditional evidence lower bound (CELBO),

$$\mathcal{L}_{\mathrm{CELBO}(\mathbf{y}|\mathbf{u})}(\theta, \phi, \psi, \xi) := \mathcal{L}_{\mathrm{ELBO}(\mathbf{y},\mathbf{u})}(\theta, \phi) - \mathcal{L}_{\mathrm{EUBO}(\mathbf{u})}(\theta, \psi, \xi). \tag{12}$$

By definition, CELBO bounds the DIG objective from below, $\mathcal{L}_{\mathbf{y}|\mathbf{u}}(\theta) \geq \mathcal{L}_{\mathrm{CELBO}(\mathbf{y}|\mathbf{u})}(\theta, \phi, \psi, \xi)$. The inequality is tight for any generative parameter $\theta$ owing to the tightness of ELBO and EUBO,[1] i.e., for all $\theta$, there exists a tuple of variational parameters $(\phi, \psi, \xi)$ such that the gap $\Delta(\theta, \phi, \psi, \xi) := \mathcal{L}_{\mathbf{y}|\mathbf{u}}(\theta) - \mathcal{L}_{\mathrm{CELBO}(\mathbf{y}|\mathbf{u})}(\theta, \phi, \psi, \xi)$ is zero. Moreover, CELBO can be unbiasedly approximated as well as ELBO and EUBO.

---

[1] ELBO is tight if $q(\mathbf{z}|\mathbf{v}, \phi) = p(\mathbf{z}|\mathbf{v}, \theta)$. EUBO is tight if $q(\mathbf{z}|\mathbf{v}, \psi) = p(\mathbf{z}|\mathbf{v}, \theta)$ and $f(\mathbf{v};\xi) = \mathcal{L}_{\mathrm{CUBO}(\mathbf{v})}(\theta, \psi) = \ln \mathcal{L}_{\mathbf{v}}(\theta)$.

---

**Algorithm 1** Variational Inference for DIG (vDIG)

---

**Input:** Data $(\mathbf{y}^{[n]}, \mathbf{u}^{[n]})$
**Output:** $\theta, \phi, \psi, \xi$
1: $(\theta, \phi, \psi, \xi) \leftarrow$ `Initialize()` // Any initialization methods can be used.
2: **repeat**
3:     Draw minibatch $B \subset [n]$
4:     $L \leftarrow \frac{1}{|B|} \sum_{i \in B} \hat{\mathcal{L}}_{\text{CELBO}(\mathbf{y}|\mathbf{u})}^i (\theta, \phi, \psi, \xi)$   // $\hat{\mathcal{L}}_{\text{CELBO}(\mathbf{y}|\mathbf{u})}$ is given by Equation 13. Use Equation 15
    instead for variance reduction.
5:     $(\theta, \phi, \psi, \xi) \leftarrow$ `Update`$((\theta, \phi, \psi, \xi), \nabla L)$ // Any gradient-based optimization methods can be used.
6: **until** converge

---

## 3.2 OPTIMIZATION ALGORITHM: vDIG

Since CELBO can be unbiasedly approximated, we may employ stochastic gradient-based optimization to maximize it. The full stochastic objective for CELBO is given by

$$\hat{\mathcal{L}}_{\text{CELBO}(\mathbf{y}|\mathbf{u})}(\theta, \phi, \psi, \xi) := \ln \frac{p(\mathbf{y}, \mathbf{u}, \mathbf{z}_\phi | \theta)}{q(\mathbf{z}_\phi | \mathbf{y}, \mathbf{u}, \phi)} - \frac{1}{\alpha} \left( \frac{p(\mathbf{u}, \mathbf{z}_\psi | \theta)}{q(\mathbf{z}_\psi | \mathbf{u}, \psi) e^{f(\mathbf{u};\xi)}} \right)^\alpha - f(\mathbf{u}; \xi) + \frac{1}{\alpha}, \quad (13)$$

where $\mathbf{z}_\phi$ and $\mathbf{z}_\psi$ are Monte-Carlo samples drawn from $q(\mathbf{z}|\mathbf{y}, \mathbf{u}, \phi)$ and $q(\mathbf{z}|\mathbf{u}, \psi)$, respectively. The gradients of $\hat{\mathcal{L}}_{\text{CELBO}(\mathbf{y}|\mathbf{u})}$ is taken with any standard automatic differentiation libraries, using the reparametrization trick (Kingma & Welling, 2013) or the REINFORCE trick (Williams, 1992).

Since the actual objective function is the summation of individual losses $\hat{\mathcal{L}}_{\text{CELBO}(\mathbf{y}|\mathbf{u})} = \hat{\mathcal{L}}_{\text{CELBO}(\mathbf{y}|\mathbf{u})}^i$ over all the instances, we may draw a minibatch of instances for each iteration. We call the resulting inference algorithm as *vDIG* (Algorithm 1).

## 3.3 VARIANCE REDUCTION FOR vDIG WITH SURROGATE PARAMETRIZATION (SP)

The boundedness of the norm of the stochastic gradient $\nabla \hat{\mathcal{L}}_{\text{CELBO}(\mathbf{y}|\mathbf{u})}(\theta, \phi, \psi, \xi)$ is crucial for the stable and fast convergence of stochastic gradient-based algorithms like Algorithm 1. However, the stochastic CELBO contains the density ratio

$$w_{\theta, \psi, \xi}(\mathbf{u}, \mathbf{z}) := \frac{p(\mathbf{u}, \mathbf{z} | \theta)}{q(\mathbf{z} | \mathbf{u}, \psi) e^{f(\mathbf{u};\xi)}} \quad (14)$$

raised to the power of $\alpha$, which is problematic as the ratio $w_{\theta, \psi, \xi}$ may have large variance and so does the gradient.

The key idea is to regularize the parameter to keep the density ratio small. Note that we have $w_{\theta, \psi, \xi}(\cdot, \cdot) \equiv 1$ whenever the objective gap is zero. That is, constraining the parameter to satisfy $\sup_{\mathbf{u}, \mathbf{z}} w_{\theta, \psi, \xi}(\mathbf{u}, \mathbf{z}) \leq 1$ does not lose the model expressibility if the variational approximation is tight.

The problem is, enforcing such constraint during optimization is intractable in general. We address this issue by introducing a surrogate parametrization. Define new parameters $\theta'$ and $\xi'$ formally[2] by

$$p(\mathbf{y}, \mathbf{u}, \mathbf{z} | \theta') := \frac{G(w_{\theta, \psi, \xi}(\mathbf{u}, \mathbf{z}))}{Z(\theta')} p(\mathbf{y}, \mathbf{u}, \mathbf{z} | \theta), \qquad f(\mathbf{u}; \xi') := f(\mathbf{u}; \xi) - \ln Z(\theta'),$$

where $G(w) := (1 \vee w)^{-1} \{1 + \alpha \ln(1 \vee w)\}^{1/\alpha}$ $(w \geq 0)$, $a \vee b := \max\{a, b\}$, and $Z(\theta')$ is the normalizing constant ensuring the mass preservation of the density under $\theta'$. We refer to the mapping $T_{\text{SP}} : (\theta, \phi, \psi, \xi) \mapsto (\theta', \phi, \psi, \xi')$ as the surrogate transform, and $G(w)$ as the gain function. The surrogate transform $T_{\text{SP}}$ and the gain function $G(w)$ are designed carefully to satisfy the following two properties.

First, it preserves the *effective parameters* of CELBO.

---

[2] Note that these parameters are conceptual objects and there is no concrete implementation for them in the final algorithm.

**Definition 1** (Effective parameters). *Let $\Omega$ be a set of CELBO parameters $(\theta, \phi, \psi, \xi)$. Then, we define the effective parameters of $\Omega$ by $\Theta_0(\Omega) := \{(\theta, \phi, \psi, \xi) \in \Omega : \Delta(\theta, \phi, \psi, \xi) = 0\}$, i.e., the set of parameters inducing tight variational approximation.*

**Proposition 2** (Effective parameter preservation). *For arbitrary $\Omega$, we have $\Theta_0(T_{\mathrm{SP}}(\Omega)) = \Theta_0(\Omega)$.*

In other words, in a sense, $T_{\mathrm{SP}}$ does not alter the original model. See Section H for extended discussion.

Second, it regularizes the growth of the highly stochastic term (Equation 14). Intuitively, the effect of regularization is quantified with the gain function $G(w)$ since it represents the ratio of stochastic term before and after the transform,

$$\frac{w_{\theta',\psi,\xi'}(\mathbf{u}, \mathbf{z})}{w_{\theta,\psi,\xi}(\mathbf{u}, \mathbf{z})} = G(w_{\theta,\psi,\xi}(\mathbf{u}, \mathbf{z})).$$

Note that $G(w)$ is no larger than one and non-increasing for $w \geq 0$, thus the stochasticity is reduced through the transform. See Figure 3 in the appendix for visualization. Consequently, $T_{\mathrm{SP}}$ guarantees the boundedness of the stochastic gradient.

**Proposition 3** (Bounded gradient). *Assume there exists $K > 0$ such that $\|\nabla \ln p(\mathbf{y}, \mathbf{u}, \mathbf{z}_\phi | \theta)\|$, $\|\nabla \ln p(\mathbf{u}, \mathbf{z}_\psi | \theta)\|$, $\|\nabla \ln q(\mathbf{z}_\phi | \mathbf{y}, \mathbf{u}, \phi)\|$, $\|\nabla \ln q(\mathbf{z}_\psi | \mathbf{u}, \psi)\|$, $\|\nabla f(\mathbf{u}; \xi)\| \leq K$. Then, $\|\nabla(\hat{\mathcal{L}}_{\mathrm{CELBO(\mathbf{y}|\mathbf{u})}} \circ T_{\mathrm{SP}})(\theta, \phi, \psi, \xi)\| \leq 9K$.*

Proposition 2 and 3 justifies optimizing the objective via the transform $T_{\mathrm{SP}}$; we get a gradient-norm bound without altering the effective parameters. The full objective function after the surrogate transform, $\hat{\mathcal{L}}_{\mathrm{CELBO\text{-}SP(\mathbf{y}|\mathbf{u})}} := \hat{\mathcal{L}}_{\mathrm{CELBO(\mathbf{y}|\mathbf{u})}} \circ T_{\mathrm{SP}}$, is given by

$$\hat{\mathcal{L}}_{\mathrm{CELBO\text{-}SP(\mathbf{y}|\mathbf{u})}}(\theta, \phi, \psi, \xi) = \ln \frac{p(\mathbf{y}, \mathbf{u}, \mathbf{z}_\phi | \theta) G(w_{\theta,\psi,\xi}(\mathbf{u}, \mathbf{z}_\phi))}{q(\mathbf{z}_\phi | \mathbf{y}, \mathbf{u}, \phi)}$$
$$- \frac{1}{\alpha} \left\{ w_{\theta,\psi,\xi}(\mathbf{u}, \mathbf{z}_\psi) G(w_{\theta,\psi,\xi}(\mathbf{u}, \mathbf{z}_\psi)) \right\}^\alpha - f(\mathbf{u}; \xi) + \frac{1}{\alpha}, \quad (15)$$

which we call the stochastic CELBO-SP. The key point is that computing the stochastic CELBO-SP does not require the computation of the normalizing constant $Z(\theta')$, which is intractable in general. This is not the case with the surrogate transformation on ELBO or EUBO alone. The CELBO-SP maximization is done by simply replacing CELBO with CELBO-SP in Algorithm 1.

### 3.4 PREDICTION ALGORITHM

Given $(\theta, \phi, \psi, \xi)$ trained by Algorithm 1, we want to compute the predictive distribution on new instances given their incomplete features $\mathbf{u}^{\mathrm{new}} := (\tilde{\mathbf{x}}^{\mathrm{new}}, \mathbf{m}^{\mathrm{new}})$. Since the conditional density $p_\theta(\mathbf{y}|\mathbf{u})$ is intractable in general, we approximate it with the Monte-Carlo method. The approximated conditional distribution is given by

$$p(\mathbf{y}|\mathbf{u}, \hat{\theta}) := \frac{1}{k_{\mathrm{pred}}} \sum_{s \in [k_{\mathrm{pred}}]} p(\mathbf{y}|\mathbf{z}_\psi^s, \theta), \qquad k_{\mathrm{pred}} \geq 1, \quad (16)$$

where $\mathbf{z}_\psi^s$ are independently drawn from $q_\psi(\mathbf{z}|\mathbf{u})$, $s \in [k_{\mathrm{pred}}]$. This procedure is justified as follows.

**Proposition 4.** *Let $p(\mathbf{y}|\mathbf{u}, \bar{\theta}) := \mathbb{E}[p(\mathbf{y}|\mathbf{u}, \hat{\theta})]$, where the expectation is taken with respect to the Monte-Carlo sampling. Then,*

$$D_{\mathrm{KL(\mathbf{y}|\mathbf{u})}}(\theta \| \bar{\theta}) \leq \frac{\alpha}{\alpha - 1} \Delta(\theta, \phi, \psi, \xi).$$

In other words, if the objective gap is small, so is the approximation error of $\bar{\theta}$, which is the limit of the actual predictor $\hat{\theta}$ with $k_{\mathrm{pred}} \to \infty$. In the experiment, we used $k_{\mathrm{pred}} = 512$.

## 4 EXPERIMENTS

In this section, we demonstrate the effectiveness of the proposed method, i.e., Algorithm 1, through numerical experiments. We first introduce a number of methods compared in the experiments in

Section 4.1. We then present the procedures and results of three experiments designed to show i) the effectiveness of the new variational approximation, ii) the superior performance of the proposed method, and iii) the robustness of the proposed method against feature corruption.

## 4.1 EXPERIMENTAL SUBJECTS

We employ three subject algorithms and two baselines in the experiment. The main subjects are all based on VAE (Kingma & Welling, 2013), a typical example of LVM, whose basic architectures are identical to each other: See the appendix (Section D) for more details.

**Generative method: VAE, VAE*.** As an example of generative approach, we employ VAE and solve missing-value imputation and regression simultaneously by regarding the target as a part of the feature, $\mathbf{x}' \leftarrow \mathbf{x} \oplus \mathbf{y}$, and reconstructing $\mathbf{x}'$ from its corrupted version, where the entries corresponding to $\mathbf{y}$ is considered as missing. In particular, we adopt the formulation of Collier et al. (2020), which comes with two variants for different missing-value processes, namely missing not at random (MNAR) and missing completely at random (MCAR). The difference between MNAR and MCAR is only in the generative model (i.e., the architecture of the decoder), represented by Equation 6 and Equation 5, respectively. We denote these variants as VAE and VAE*, respectively.

To train it to be able to reconstruct $\mathbf{y}$, we double the training dataset with masking (and not masking) the target $\mathbf{y}$. Note that the objective function is not exactly aligned with the prediction task because of the imputation-based formulation.

**Discriminative method: CVAE.** As an example of discriminative approach, we employ the conditional VAE (CVAE) proposed by Kingma et al. (2014); Sohn et al. (2015). In our setting, the conditional variables consist of the corrupted feature $\tilde{\mathbf{x}}$ and the missingness indicator $\mathbf{m}$. In practice, these vectors are concatenated before fed into neural networks. The generative model of CVAE in this setting is given by $p(\mathbf{y}|\mathbf{u}, \theta) := \int p(\mathbf{y}, \mathbf{z}|\mathbf{u}, \theta)\, \mathrm{d}\mathbf{z}$, while the decoder represents the density in the integral and the encoder is used to approximate the posterior $p(\mathbf{z}|\mathbf{y}, \mathbf{u}, \theta)$. Note that the resulting model is not informed with the context of $\mathbf{m}$ as VAE and VAE* are,[3] but the objective is (a variational approximation of) the conditional evidence, which is aligned with the prediction task.

**Proposed method: DVAE, DVAE*.** We call the instantiation of vDIG on VAE as the discriminative VAE (DVAE). We implement MNAR (Equation 6) and MCAR (Equation 5) variants of DVAE as with VAE, respectively denoted by DVAE and DVAE*. The difference with VAE and its variant is that we have additional encoders corresponding to $\psi$ and $\xi$, while the decoder $\theta$ and the encoder $\phi$ are common. See Section D for more details. We choose $\alpha = 2$ for the parameter of the Rényi divergence.

**Baseline methods: Simple, MICE.** As the first baseline method, we employ a simple fully-connected feed-forward neural network (FCFNN) with the same architecture with the encoders of the above methods, except its output is considered as a distribution of the target $\mathbf{y}$ instead of $\mathbf{z}$. Although the output distribution is restricted to a Gaussian, it requires no variational approximation and hence the conditional likelihood $\ln p_\theta(\mathbf{y}|\mathbf{u})$ is exactly maximized. We call this Simple. The second baseline is the method of the multiple imputation by chained equations (MICE) (Azur et al., 2011), which is a classic approach in statistics to deal with missing values. In particular, we employ the Bayesian ridge regression for the missing-value imputation task and the FCFNN of the first baseline for the subsequent regression task.

## 4.2 EXPERIMENTAL PROCEDURE AND RESULTS

The organization of the experiments is three-fold. First, in Section 4.2.1, we check the effectiveness of our variational approximation techniques, namely EUBO and the surrogate parameterization. Second, in Section 4.2.2, we compared the proposed method(s) with the existing methods in terms of the predictive performance. Finally, in Section 4.2.3, we examine the robustness of these methods against the change in the missing-value ratio. See also the appendix for the details of the experimental settings, including the information of the datasets.

---

[3]Because of this, there is no MCAR variant for CVAE.

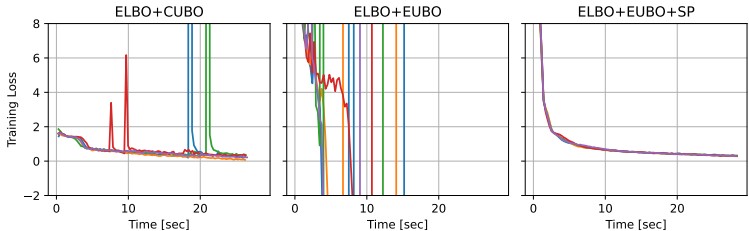

Figure 1: Each of three panels corresponds to a different variational approximation. Lines in each panel denote the training objectives during 5 independent training runs.

|  | AQ-CO | AQ-NMHC | AQ-NOx | Boston | Diabetes | YearPred | Total |
|---|---|---|---|---|---|---|---|
| CVAE | **0.41** (0.02) | 5.19 (0.14) | 5.33 (0.02) | 3.01 (0.18) | 5.46 (0.03) | 3.49 (0.06) | 3.82 (0.04) |
| DVAE | 0.45 (0.03) | **5.16** (0.10) | **5.26** (0.02) | 2.91 (0.23) | 5.44 (0.08) | 3.36 (0.04) | **3.76** (0.05) |
| DVAE* | 0.44 (0.03) | **5.16** (0.11) | 5.27 (0.03) | 2.92 (0.21) | **5.42** (0.07) | **3.35** (0.02) | **3.76** (0.04) |
| Simple | 0.49 (0.03) | 5.47 (0.11) | 5.40 (0.02) | 3.13 (0.18) | 5.47 (0.04) | 3.62 (0.04) | 3.93 (0.04) |
| MICE | 0.46 (0.06) | 5.34 (0.23) | 5.38 (0.02) | 2.94 (0.10) | 5.46 (0.03) | 3.61 (0.08) | 3.87 (0.05) |
| VAE | 0.47 (0.01) | 5.21 (0.08) | 5.49 (0.06) | 2.95 (0.23) | 5.48 (0.10) | 3.59 (0.02) | 3.87 (0.04) |
| VAE* | 0.46 (0.02) | 5.20 (0.09) | 5.47 (0.05) | **2.81** (0.20) | 5.46 (0.05) | 3.58 (0.03) | 3.83 (0.04) |

Table 1: Test per-sample cross entropies of different algorithms for different datasets. The numbers represent the mean (and standard deviation in parenthesis) of five independent runs for each configuration. For each column, the best score is indicated with bold face.

### 4.2.1 EFFECT OF NEW VARIATIONAL APPROXIMATION

**Procedure.** We compare the stability of optimization of DVAE with three different variational approximation, namely, ELBO+CUBO, ELBO+EUBO and ELBO+EUBO+SP. In all cases, the Monte-Carlo approximation of ELBO is used to compute the joint evidence $\mathcal{L}_{\mathbf{y},\mathbf{u}}(\theta)$. The marginal evidence $\mathcal{L}_{\mathbf{u}}(\theta)$ is computed with the Monte-Carlo approximation for both CUBO (albeit biased) and EUBO. Only ELBO+EUBO+SP employs the surrogate parametrization.

**Result.** Figure 1 shows the training processes of DVAE with the AQ-CO dataset from UCI Machine Learning Repository (see the appendix). It is seen that ELBO+EUBO+SP is most stable in the optimization. Also note that the objective of ELBO+CUBO may be negatively biased (recall the logarithm wrapping the expectation in Equation 10), but we cannot know how much.

### 4.2.2 PREDICTIVE PERFORMANCE

**Procedure.** We apply seven different algorithms in Section 4.1 to six different regression tasks from UCI Machine Learning Repository, summarized in Table 3 (in the appendix). We drop the entries of feature completely at random with probability $p = 0.1$. In each configuration, we iterate the same procedure with five different random seed values.

**Result.** Table 1 shows the cross entropy, i.e., $-\ln p_\theta(\mathbf{y}|\mathbf{u})$, averaged over the test splits. Overall, both DVAE and DVAE* almost always perform best or at least comparable to the best scores within one standard deviation, indicating the effectiveness the DIG method applied VAE. In particular, it is more clear that they outperform the others when averaged over all the datasets ('Total' column). Moreover, DVAE* is slightly better than DVAE as expected because the way we drop the feature entries is MCAR.

### 4.2.3 ROBUSTNESS AGAINST FEATURE CORRUPTION

**Procedure.** We take the result of Section 4.2.2 and examine how the missing-value ratio $p$ affects predictive performance. For each method, we calculate the difference between the average cross entropy with $p = 0.1$ and $p = 0.5$.

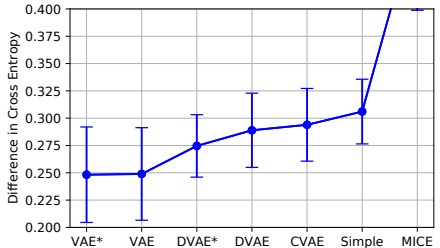

Figure 2: The average performance loss when the missing-value ratio is changed from $p = .1$ to $p = .5$. Lower values imply more robustness against feature incompletion. The error bars represent the estimated standard deviation of the expectations.

**Result.** The difference is visualized in Figure 2. The most robust algorithms are VAE and VAE*, while DVAE* and DVAE are the second and the third most robust algorithms. This matches our expectation because both VAE and VAE* are informed with the context of $\mathbf{m}$ and implicitly regularized through missing-value imputation task. Similarly, DVAE and DVAE* are also informed with the meaning of $\mathbf{m}$, but they are not subject to the implicit regularization. In particular, DVAE* is more robust than DVAE as the MCAR model is accurate (in this setting) and more informative than the MNAR model.

## 5 RELATED WORK

In this section, we review existing variational approximations bounding the log-integrals from *above*. We also discuss the relationship with the existing studies on missing data handling in the appendix.

The $\chi$-variational upper bound (CUBO) (Dieng et al., 2017) has been proposed to estimate such upper bounds, utilizing the $\alpha$-Rényi divergence (equivalently, $\chi$-divergence). However, as Pradier et al. (2019) pointed out, the resulting estimate of CUBO is biased and in some cases numerically unstable[4]. This phenomenon has been also confirmed in our experiment (Section 4.2.1). Another upper bound has been proposed by (Ji & Shen, 2019) with the reverse KL divergence instead of the Rényi divergence, but it is also biased and not guaranteed to be an upper bound. We also find that the parsimonious upper bound (Mattei & Frellsen, 2018) is suitable in terms of the upper-bound guarantee, while it induces a min-max form during the upper bound minimization procedure, which implies the convergence property of the algorithm could be tricky.

The proposed upper bound, EUBO, is closely related to the one studied by Kuleshov & Ermon (2017), which is designed for estimating partition functions. Although their method also suffers from high variance, they took different approach to reduce it. In particular, they adaptively reduced the learning rate of the variational parameter $\psi$. However, as the authors noted, this does not solve the problem enough for scaling to large datasets.

## 6 CONCLUSION

We have proposed a novel algorithm to perform discriminative training with incomplete features for generative models, which is derived on the basis of a newly introduced variational approximation and parameter transformation. The effectiveness of the proposed method has been confirmed in terms of the stability of the new upper bound and the predictive performance and robustness of the resulting algorithm.

Possible directions of future work include the application of the surrogate transform technique to other contexts than feature incompletion.

---

[4]Another work (Lopez et al., 2020) reported that there is a case CUBO works just fine, so its seems a problem dependent phenomenon.

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

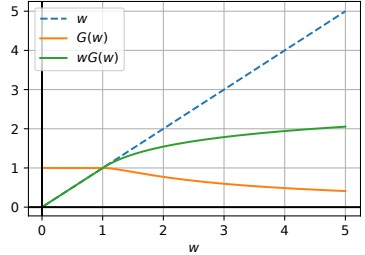

Figure 3: The gain function $G(w)$ with $\alpha = 2$ (the orange line). The green line shows the size of the stochastic term $w_{\theta,\psi,\xi}(\mathbf{u}, \mathbf{z})$ after the transformation, whereas the blue dashed line shows the original size.

## A PROOFS

### A.1 PROOF OF PROPOSITION 2

*Proof.* $\Theta_0(T_{\text{SP}}(\Omega)) \supset \Theta_0(\Omega)$ is trivial as $\Delta(\theta, \phi, \psi, \xi) = 0$ implies $w_{\theta,\psi,\xi}(\cdot, \cdot) \equiv 1$, which implies $(\theta, \phi, \psi, \xi)$ is a fixed point of $T_{\text{SP}}$, hence $(\theta, \phi, \psi, \xi) \in T_{\text{SP}}(\Omega)$. The other direction is shown as follows. Assume $\Delta(\theta', \phi, \psi, \xi') = 0$, where $(\theta', \phi, \psi, \xi') = T_{\text{SP}}(\theta, \phi, \psi, \xi)$ and $(\theta, \phi, \psi, \xi) \in \Omega$. Then, we have

$$1 = w_{\theta',\psi,\xi'}(\mathbf{u}, \mathbf{z}) = w_{\theta,\psi,\xi}(\mathbf{u}, \mathbf{z})G(w_{\theta,\psi,\xi}(\mathbf{u}, \mathbf{z}))$$

for all $\mathbf{u}$ and $\mathbf{z}$. This implies $w_{\theta,\psi,\xi}(\cdot, \cdot) \equiv 1$ as $wG(w) = 1 \Leftrightarrow w = 1$. Therefore, $(\theta, \phi, \psi, \xi)$ is a fixed point of $T_{\text{SP}}$ and $(\theta', \phi, \psi, \xi') \in \Omega$. $\square$

### A.2 PROOF OF PROPOSITION 3

*Proof.* Let $g := \nabla(\hat{\mathcal{L}}_{\text{CELBO}(\mathbf{y}|\mathbf{u})} \circ T_{\text{SP}})(\theta, \phi, \psi, \xi)$ and observe

$$g = \nabla \ln \frac{p(\mathbf{y}, \mathbf{u}, \mathbf{z}_\phi|\theta)G(w)}{q(\mathbf{z}_\phi|\mathbf{y}, \mathbf{u}, \phi)} - \frac{\nabla}{\alpha}(wG(w))^\alpha - \nabla f(\mathbf{u}; \xi)$$

$$= \nabla \ln \frac{p(\mathbf{y}, \mathbf{u}, \mathbf{z}_\phi|\theta)}{q(\mathbf{z}_\phi|\mathbf{y}, \mathbf{u}, \phi)} - \nabla f(\mathbf{u}; \xi) + \nabla H(\ln w)$$

$$= \nabla \ln \frac{p(\mathbf{y}, \mathbf{u}, \mathbf{z}_\phi|\theta)}{q(\mathbf{z}_\phi|\mathbf{y}, \mathbf{u}, \phi)} - \nabla f(\mathbf{u}; \xi) + H'(\ln w)\nabla \ln w$$

where $w := w_{\theta,\psi,\xi}(\mathbf{u}, \mathbf{z})$, $H(t) := \ln G(e^t) - (e^t G(e^t))^\alpha/\alpha$, and $H'(t)$ denotes the derivative of $H(t)$. Thus, we have

$$\|g\| \leq |H'(\ln w)| \|\nabla \ln w\| + 3K \leq 9K$$

as $|H'(t)| \leq 2$ and $\|\nabla \ln w\| \leq 3K$. $\square$

### A.3 PROOF OF PROPOSITION 4

*Proof.* According to the information processing inequality, we have $D_{\text{KL}(\mathbf{y}|\mathbf{u})}(\theta\|\bar{\theta}) \leq D_{\text{KL}(\mathbf{z}|\mathbf{u})}(\theta\|\psi)$. Moreover, by the construction of $\mathcal{L}_{\text{MEUBO}}$, we have $(1 - \alpha^{-1})D_{\alpha(\mathbf{z}|\mathbf{u})}(\theta\|\psi) \leq \Delta(\theta, \phi, \psi, \xi)$. The desired result is seen by combining these two inequalities with the fact that the $\alpha$-Rényi divergence dominates the KL divergence for all $\alpha > 1$. $\square$

## B EXPERIMENTAL CONFIGURATIONS

**Computing infrastructure.** The computing infrastructure used in the experiments is summarized in Table 2.

| CPU | RAM | GPU | PyTorch |
|---|---|---|---|
| Intel(R) Core(TM) i7-6700K CPU @ 4.00GHz | 64GB | NVIDIA TITAN X | 1.9.0 |

Table 2: Summary of computing infrastructure.

**Preprocessing.** All datasets used in the experiment are standardized before fed into algorithms so that the variables of the feature and the target have zero mean and unit empirical variance on the training split. More precisely, we apply the following transformation

$$\tilde{\mathbf{x}}^i \leftarrow (1 - \mathbf{m}^i) \odot (\tilde{\mathbf{x}}^i - \hat{\mu}_x) \oslash \hat{\sigma}_x, \qquad \mathbf{y}^i \leftarrow (\mathbf{y}^i - \hat{\mu}_y) \oslash \hat{\sigma}_y, \qquad i \in [n],$$

where $\hat{\mu}_{x,j} := \frac{\sum_{i=1}^n (1-m_j^i) x_j^i}{\sum_{i=1}^n (1-m_j^i)}$ and $\hat{\sigma}_{x,j}^{\odot 2} := \frac{\sum_{i=1}^n (1-m_j^i)(x_j^i - \hat{\mu}_{x,j})^{\odot 2}}{\sum_{i=1}^n (1-m_j^i)}$ for $j \in [d_x]$, and $\hat{\mu}_y := \frac{1}{n} \sum_{i=1}^n y^i$ and $\hat{\sigma}_y^{\odot 2} := \frac{1}{n} \sum_{i=1}^n (y^i - \hat{\mu}_y)^{\odot 2}$. Here, $\odot$ and $\oslash$ denotes the elementwise multiplication and division of vectors, respectively. Note that we refill the missing entries of $\tilde{\mathbf{x}}^i$ with zero after the affine transformation, which is corresponding to filling them with the means in the original dataset. In the prediction step, the input feature $\tilde{\mathbf{x}}$ undergoes the same transformation, $\tilde{\mathbf{x}}^i \leftarrow (1 - \mathbf{m}) \odot (\tilde{\mathbf{x}} - \hat{\mu}_x) \oslash \hat{\sigma}_y$, and the predictive distribution $p(\mathbf{y}|\mathbf{u}, \theta)$ for the standardized target is made based on it. Then, the distribution is transformed inversely to predict the non-standardized target, $p(\mathbf{y}|\mathbf{u}, \theta) \leftarrow (\prod_{j=1}^{d_y} \hat{\sigma}_{j,y}^{-1}) \cdot p((\mathbf{y} - \hat{\mu}_y) \oslash \hat{\sigma}_y | \mathbf{u}, \theta)$.

**Validation split.** We randomly hold 20% of the training split out for validation. Algorithms receive as the input only the remaining 80%.

**Initialization.** All the parameters involved in the experiments are contained in the `Linear` module of Pytorch and initialized with its default initialization method, i.e., independently subject to the uniform distribution on the interval $[-\frac{1}{\sqrt{k}}, \frac{1}{\sqrt{k}}]$, where $k$ is the number of input variables for the respective modules.

**First-order optimizer.** In all the experiments, we employ the AdaBelief optimizer (Zhuang et al., 2020) with its default setting,[5] which is recently proposed as an improved alternative for the Adam optimizer (Kingma & Ba, 2014). The size of minibatch is always taken to be 512. In particular, for sampling-based methods (i.e., VAE, VAE*, CVAE, DVAE, DVAE*), the stochastic gradient is computed with the reparametrization trick.

**Number of Iterations.** We iterate the loop 2000 times for each configuration, i.e., $2000 \times 512$ samples are seen in one run irrespective to the data size.

**Evaluation.** In the above iteration, we save the models at 20 predefined points $t \in \{0, 1, 2, 3, 5, 8, 13, 19, 29, 43, 64, 94, 138, 203, 298, 436, 638, 934, 1366, 1998\}$, which are (approximately) equally spaced in the log scale. Finally, the best model among these 20 checkpoints is chosen with respect to the cross entropy on the validation split and evaluated on the test split. If the subject algorithm is based on LVM, then we sample $k_{\text{pred}} = 512$ instances from the posterior distribution $q(\mathbf{z}|\mathbf{u}, \psi)$ and make prediction with Equation 16.

## C  DATASETS

All the datasets used in the experiments are taken from UCI Machine Learning Repository. AQ-CO, AQ-NMHC and AQ-NOx are taken from the AirQuality dataset (De Vito et al., 2008) and corresponding to different targets denoted by their suffixes. YearPred is a part of the YearPrecitonMSD dataset (Bertin-Mahieux, 2011), where, to accommodates the fast iteration of the experiments, the number of records is restricted to 10,000 by separate random sampling for training (8000 records) and test (2000 records) splits. Boston and Diabetes are respectively taken from the dataset of the same names. The basic statistics are summarized in Table 3.

---

[5]See https://github.com/juntang-zhuang/Adabelief-Optimizer/tree/update_0.2.0

|       | AQ-CO | AQ-NMHC | AQ-NOx | Boston | Diabetes | YearPred* |
|-------|-------|---------|--------|--------|----------|-----------|
| $n$   | 6139  | 731     | 6174   | 404    | 353      | 10000     |
| $d$   | 10    | 10      | 10     | 13     | 10       | 90        |

Table 3: Summary statistics of datasets. '$*$' indicates subsampled dataset.

## D  METHODS COMPARED

In this section, we show the detailed architecture of the methods compared in the experiments.

Both the encoder and the decoder are fully-connected feed-forward neural networks (FCFNN) with single hidden layer of width 256 and ReLU activation that take a vector input $\mathbf{t} \in \mathbb{R}^k$, $k \geq 1$, and emit a diagonal Gaussian distribution depending on $\mathbf{t}$. To facilitate the specification of the architectures, we first define some basic building blocks.

Let every distinct occurrence of $\texttt{Linear}_d(\cdot)$ denote a distinct linear layer with the output dimension $d \geq 1$ with its own parameters $(W, \mathbf{b})$ such that $\texttt{Linear}_d(\mathbf{t}) = W\mathbf{t} + \mathbf{b}$ with a weight matrix $W \in \mathbb{R}^{d \times *}$ and an offset vector $\mathbf{b} \in \mathbb{R}^d$ for $\mathbf{t} \in \mathbb{R}^*$. The Gaussian layer is defined by the composition of the Gaussian density function and the linear layers,

$$\texttt{Gauss}_d(\mathbf{t}|\mathbf{v}) := \mathcal{N}_d[\mathbf{t} \,|\, \texttt{Linear}_d(\mathbf{v}), \epsilon + \ln(1 + \texttt{Linear}_d(\mathbf{v}))]$$

where $\epsilon := 10^{-8}$ is a small constant and $\mathcal{N}_d[\mathbf{t}|\mathbf{v}, \mathbf{s}]$ denotes the $d$-product of the probability density functions of the Gaussian distributions with mean $\mathbf{v}_j$ and standard deviation $\mathbf{s}_j$ evaluated at $\mathbf{t}_j$, $j \in [d]$, for $\mathbf{t}, \mathbf{v} \in \mathbb{R}^d$ and $\mathbf{s} \in \mathbb{R}^d_{>0}$. We also denote the ReLU activation function by $\texttt{ReLU}(\mathbf{t}) := \max\{0, \mathbf{t}\}$. Finally, we define the composition of a Gaussian layer and a Linear layer with $k$ hidden neurons and ReLU activation function by

$$\texttt{Gauss-NN}_{d,k}(\mathbf{t}|\mathbf{v}) := \texttt{Gauss}_d(\mathbf{t}|(\texttt{ReLU} \circ \texttt{Linear}_k)(\mathbf{v})).$$

Now, we are ready to describe the architecture specifications.

**Simple.** The objective of the Simple baseline is given by

$$-\ln p(\mathbf{y}|\mathbf{u}, \theta) := -\ln \texttt{Gauss-NN}_{d_y, 256}(\mathbf{y}|\tilde{\mathbf{x}} \oplus \mathbf{m}),$$

where $\theta$ denotes all the parameter implicitly involved in the RHS.

**MICE.** MICE is implemented with $\texttt{IterativeImputer}$ from $\texttt{scikit-learn}$ (ver. 0.24.2) with its default argument except with the burn-in period changed from 10 to 40. After the imputation, only the final imputation result is passed to the Simple baseline to reduce the computation time.

**VAE, VAE*.** The objective of VAE is given slightly modifying Collier et al. (2020). The difference is that the mask vector $\mathbf{m}$ is modeled as a conditional variable rather than a random variable. The reason of this modification is because we are not interested in the likelihood of $\mathbf{m}$ and conditioning on such variables simplifies the resulting model. Namely,

$$\hat{\mathcal{L}}_{\text{VAE}}(\theta, \phi) := -\ln p(\mathbf{y}, \mathbf{u}|\mathbf{z}, \theta) - \ln \mathcal{N}_{10}[\mathbf{z}; 0, 1] + \ln q(\mathbf{z}|\mathbf{y}, \mathbf{u}, \phi),$$

where

$$p(\mathbf{y}, \mathbf{u}|\mathbf{z}, \theta) := p(\mathbf{m}) \cdot \texttt{Gauss}_{d_y}(\mathbf{y}|\mathbf{v}) \cdot \prod_{j \in [d_x]: \mathbf{m}_j = 0} \texttt{Gauss}_1(\tilde{\mathbf{x}}_j|\mathbf{v}),$$

$$q(\mathbf{z}|\mathbf{y}, \mathbf{u}, \phi) := \texttt{Gauss-NN}_{10, 256}(\mathbf{z}|\mathbf{y} \oplus \tilde{\mathbf{x}} \oplus \mathbf{m}),$$

$\mathbf{v} := (\texttt{ReLU} \circ \texttt{Linear}_{256})(\mathbf{z} \oplus \mathbf{m})$ and $\mathbf{z} \sim q(\mathbf{z}|\tilde{\mathbf{x}}, \mathbf{m}, \phi)$. For VAE*, replace $\mathbf{v}$ with $\mathbf{v}^* := (\texttt{ReLU} \circ \texttt{Linear}_{256})(\mathbf{z})$. Here, $p(\mathbf{m})$ is an arbitrary density function of $\mathbf{m}$, which is constant with respect to the parameters $(\theta, \phi)$ and ignored in the gradient computation.

|  | AQ-CO | AQ-NMHC | AQ-NOx | Boston | Diabetes | YearPred | Total |
|---|---|---|---|---|---|---|---|
| CVAE | **0.76** (0.04) | 5.62 (0.13) | 5.82 (0.05) | 3.32 (0.21) | 5.56 (0.05) | 3.57 (0.10) | 4.11 (0.05) |
| DVAE | **0.76** (0.03) | 5.53 (0.08) | **5.74** (0.03) | 3.31 (0.24) | 5.53 (0.02) | 3.44 (0.02) | 4.05 (0.04) |
| DVAE* | **0.76** (0.04) | 5.54 (0.08) | 5.75 (0.03) | **3.23** (0.15) | **5.50** (0.02) | **3.43** (0.02) | **4.03** (0.03) |
| Simple | 0.87 (0.04) | 5.87 (0.08) | 5.96 (0.07) | 3.43 (0.21) | 5.57 (0.02) | 3.71 (0.03) | 4.23 (0.04) |
| MICE | 0.95 (0.06) | 6.07 (0.70) | 6.01 (0.07) | 3.80 (0.59) | 5.61 (0.06) | 3.68 (0.02) | 4.35 (0.15) |
| VAE | 0.80 (0.03) | 5.53 (0.07) | 5.88 (0.03) | 3.37 (0.37) | 5.57 (0.03) | 3.55 (0.03) | 4.12 (0.06) |
| VAE* | 0.81 (0.04) | **5.49** (0.10) | 5.87 (0.04) | 3.28 (0.40) | **5.50** (0.04) | 3.53 (0.04) | 4.08 (0.07) |

Table 4: The result of the same experiment of Section 4.2.2 except the missing-value ratio is $p = 0.5$ instead of $p = 0.1$.

**CVAE.** The objective of CVAE is given according to Sohn et al. (2015),

$$\hat{\mathcal{L}}_{\text{CVAE}}(\theta, \phi) := -\ln p(\mathbf{y}|\mathbf{z}, \mathbf{u}, \theta) - \ln \mathcal{N}_{10}[\mathbf{z}; 0, 1] + \ln q(\mathbf{z}|\mathbf{y}, \mathbf{u}, \phi),$$

where

$$p(\mathbf{y}|\mathbf{z}, \mathbf{u}, \theta) := \texttt{Gauss-NN}_{d_y, 256}(\mathbf{y}|\mathbf{z} \oplus \tilde{\mathbf{x}} \oplus \mathbf{m}),$$
$$q(\mathbf{z}|\mathbf{y}, \mathbf{u}, \phi) := \texttt{Gauss-NN}_{10, 256}(\mathbf{z}|\mathbf{y} \oplus \tilde{\mathbf{x}} \oplus \mathbf{m}),$$

and $\mathbf{z} \sim q(\mathbf{z}|\mathbf{y}, \mathbf{u}, \phi)$.

**DVAE, DVAE*.** The objective of DVAE is given by Equation 13, where

$$p(\mathbf{y}, \mathbf{u}|\mathbf{z}, \theta) := p(\mathbf{m}) \cdot \texttt{Gauss}_{d_y}(\mathbf{y}|\mathbf{v}) \cdot \prod_{j \in [d_x]: \mathbf{m}_j = 0} \texttt{Gauss}_1(\tilde{\mathbf{x}}_j|\mathbf{v}),$$

$$q(\mathbf{z}|\mathbf{y}, \mathbf{u}, \phi) := \texttt{Gauss}_{10}(\mathbf{z}|g_{256}(\mathbf{y}, \mathbf{0}_{d_y}, \tilde{\mathbf{x}}, \mathbf{m}; \omega)),$$
$$q(\mathbf{z}|\mathbf{u}, \psi) := \texttt{Gauss}_{10}(\mathbf{z}|g_{256}(\mathbf{0}_{d_y}, \mathbf{1}_{d_y}, \tilde{\mathbf{x}}, \mathbf{m}; \omega)),$$
$$f(\mathbf{u}; \xi) := \ln p(\mathbf{m}) + \texttt{Linear}_1(g_{256}(\mathbf{0}_{d_y}, \mathbf{1}_{d_y}, \tilde{\mathbf{x}}, \mathbf{m}; \omega)),$$

$g_k(\mathbf{t}^{(1)}, \mathbf{t}^{(2)}, \mathbf{t}^{(3)}, \mathbf{t}^{(4)}; \omega) := (\texttt{ReLU} \circ \texttt{Linear}_k)(\mathbf{t}^{(1)} \oplus \mathbf{t}^{(2)} \oplus \mathbf{t}^{(3)} \oplus \mathbf{t}^{(4)})$ for $\mathbf{t}^{(1)}, \mathbf{t}^{(2)} \in \mathbb{R}^{d_y}$ and $\mathbf{t}^{(3)}, \mathbf{t}^{(4)} \in \mathbb{R}^{d_x}$, $\omega$ denotes the shared parameter of $\phi, \psi, \xi$, and $\mathbf{v} := (\texttt{ReLU} \circ \texttt{Linear}_{256})(\mathbf{z} \oplus \mathbf{m})$. For DVAE*, replace $\mathbf{v}$ with $\mathbf{v}^* := (\texttt{ReLU} \circ \texttt{Linear}_{256})(\mathbf{z})$. Here, $p(\mathbf{m})$ is an arbitrary density function of $\mathbf{m}$, which cancels out in the final objective of CELBO or CELBO-SP.

# E  ADDITIONAL RESULTS

We show in Table 4 the results of the same experiment as in Section 4.2.2 except with the increased missing-value ratio $p = 0.5$.

# F  ADDITIONAL DISCUSSION ON RELATED WORK: MISSING-DATA HANDLING

Arguably the most classic approach to the missing value problem is the two-step approach, also known as the imputation method (Enders, 2010). This category includes traditional listwise or pair-wise deletion methods, single imputation methods and multiple imputation methods. The key feature of this approach is that one processes the incomplete features to get estimates of complete ones $\hat{\mathbf{x}} \approx \mathbf{x}$ in the first step and then performs prediction based on $\hat{\mathbf{x}}$. The score-based methods (Jaakkola & Haussler, 1999; Holub et al., 2005; Perina et al., 2009; Śmieja et al., 2018) can be considered as a generalization of the two-step approach developed in the machine learning literature. With a score-based method, one first learns the generative distribution $p_\theta(\mathbf{x})$, which is used to extract some information called *score*, $s$. The score is then fed to predictive models $p_\theta(\mathbf{y}|s)$. Several drawbacks stem from the two-step nature of these methods. In case of the imputation method, the predictor loses the information whether each element of $\hat{\mathbf{x}}$ is original or imputed. This makes it difficult to estimate the uncertainty resulted from the feature corruption. Moreover, even though the uncertainty

| Approach | Modeling | Objective | Missingness-Aware | Objective Alignment |
|---|---|---|---|---|
| Two-step | $s(\mathbf{u}), p_\theta(\mathbf{y}|s)$ | $\ln p_\theta(\mathbf{y}|s(\mathbf{u}))$ | -/+ | - |
| Generative | $p_\theta(\mathbf{y}, \mathbf{u})$ | $\ln p_\theta(\mathbf{y}, \mathbf{u})$ | + | - |
| Discriminative | $p_\theta(\mathbf{y}|\mathbf{u})$ | $\ln p_\theta(\mathbf{y}|\mathbf{u})$ | - | + |
| DIG | $p_\theta(\mathbf{y}, \mathbf{u})$ | $\ln p_\theta(\mathbf{y}|\mathbf{u})$ | + | + |

Table 5: Summarized comparison of related work.

problem can be addressed with the score-based method, learning good score representations requires solving optimization problems possibly irrelevant to the original prediction problem and thus it may compromise the performance.

Another category of missing feature handling fully utilizes generative models. One of such method is referred to as the full information maximum likelihood methods (see also Secion 4, Enders (2010)), where the joint complete-data distribution $p_\theta(\mathbf{y}, \mathbf{x})$ is explicitly modeled and marginalized over the corrupted elements of features to obtain the objective $\ln p_\theta(\mathbf{y}, \mathbf{u})$. This approach may suffer from unnecessarily performance degradation as in the score-based approach, since the objective contains the generative term of $\mathbf{u}$, $\ln p_\theta(\mathbf{y}, \mathbf{u}) = \ln p_\theta(\mathbf{y}|\mathbf{u}) + \ln p_\theta(\mathbf{u})$.

The third approach is the discriminative approach. Specifically, tree-based models such as gradient boosting trees are able to naturally handle incomplete features (Twala et al., 2008). Moreover, it is also recommended in (Kuhn et al., 2013) to encode missingness as a distinct feature, i.e., treat the concatenation of the corrupted feature and the mask vectors, $\tilde{\mathbf{x}} \oplus \mathbf{m}$, as the input to the predictive models. The advantage of this approach is that the resulting objective function is coherent with the goal of predictive risk minimization, i.e., there is no generative term unlike in the two-step and generative approach. However, there is no trivial way to inform the model that $\mathbf{m}$ actually indicates missing features. Therefore it may take extra samples to learn the meaning of $\mathbf{m}$ by itself.

Finally, the DIG approach can be thought of as a hybrid of the generative and discriminative approaches. With DIG, the data is modeled with joint distribution $p_\theta(\mathbf{y}, \mathbf{u})$, but the learning objective is the conditional evidence $\ln p_\theta(\mathbf{y}|\mathbf{u})$, which is computed from Bayes' rule. Therefore, it naturally incorporates the information of missingness and is directly trained to maximize the predictive performance. The first application of the DIG strategy in the context of incomplete feature is Ghahramani & Jordan (1994), which was followed by Smola et al. (2005) with a kernel-based generalization. It is crucial in their results that the complete-data model $p_\theta(\mathbf{y}, \mathbf{x})$ is a exponential family so that the objective function is optimized with the EM algorithm (Dempster et al., 1977). A tractable approximation algorithm for the case of Gaussian processes is derived by Pacheco et al. (2014) with the combination of the variational inference (Jordan et al., 1999) and the expectation propagation (Minka, 2001) framework. As opposed to their method, our focus is on black-box variational inference algorithm applicable to a variety of models.

See Table 5 for the summary of the comparison.

## G  MODELING THE MAR PROCESS

In this section, we give a method of modeling the third type of the feature-corruption processes, the MAR process. Let $\mathcal{O} \subset [d_x]$ be an index set on which the feature is always observed (i.e., $\mathbf{m}_j = 0$ for all $j \in \mathcal{O}$) and define $\tilde{\mathbf{x}}_\mathcal{O} := \{\tilde{\mathbf{x}}_j : j \in \mathcal{O}\}$. Then, the MAR process is modeled with

$$p(\mathbf{y}, \mathbf{u}, \mathbf{z}|\theta) = p(\tilde{\mathbf{x}}_\mathcal{O}, \mathbf{m}) \cdot p(\mathbf{z}) \cdot p(\mathbf{y}|\mathbf{z}, \tilde{\mathbf{x}}_\mathcal{O}, \theta) \cdot \prod_{j \in [d_x] \setminus \mathcal{O}} \{p(\tilde{\mathbf{x}}_j|\mathbf{z}, \tilde{\mathbf{x}}_\mathcal{O}, \theta)\}^{1-\mathbf{m}_j}, \quad (17)$$

where $p(\tilde{\mathbf{x}}_\mathcal{O}, \mathbf{m})$ will cancel out as well as $p(\mathbf{m})$ in the MNAR and MCAR processes.

## H  ADDITIONAL JUSTIFICATION AND LIMITATION OF THE SURROGATE TRANSFORM

We present another justification of the CELBO-SP maximization (i.e., optimization of Equation 15) as an alternative to the CELBO maximization (i.e., optimization of Equation 13). Recall that

CELBO-SP is derived by the surrogate transform $T_{\text{SP}}$ and Proposition 2 shows a desirable property of $T_{\text{SP}}$ viewing it as an operator acting on the *parameters*. In this section, we show another desirable property of $T_{\text{SP}}$ viewing it as an operator acting on the *objective function*. More specifically, we view $T_{\text{SP}}$ as a mapping from $\hat{\mathcal{L}}_{\text{CELBO}(\mathbf{y}|\mathbf{u})}$ to $\hat{\mathcal{L}}_{\text{CELBO-SP}(\mathbf{y}|\mathbf{u})} = \hat{\mathcal{L}}_{\text{CELBO}(\mathbf{y}|\mathbf{u})} \circ T_{\text{SP}}$ and discuss the relationship of these objective functions.

Let $\zeta := (\phi, \psi, \xi)$ denote the tuple of the variational parameters for brevity and $\Omega$ be the set of the parameters $(\theta, \zeta)$ on which we perform the optimization. Let $\theta^*$ denote the true generative parameter of $(\mathbf{y}, \mathbf{u})$, which is not necessarily contained in $\Omega$. Finally, define the discrepancy function of $(\theta, \zeta) \in \Omega$ with respect to $\theta^*$ by

$$\delta(\theta^* \| \theta, \zeta) := D_{\text{KL}(\mathbf{y}|\mathbf{u})}(\theta^* \| \theta) + \Delta(\theta, \zeta),$$

where $D_{\text{KL}(\mathbf{y}|\mathbf{u})}(\theta^* \| \theta) := \mathbb{E}_{(\mathbf{y},\mathbf{u}) \sim \theta^*}[\ln \frac{p(\mathbf{y}|\mathbf{u},\theta^*)}{p(\mathbf{y}|\mathbf{u},\theta)}]$ is the conditional Kullback–Leibler divergence of $\theta^*$ and $\theta$ and $\Delta(\theta, \zeta) = \Delta(\theta, \phi, \psi, \xi)$ is the gap function defined just after Equation 12. Note that it is nonnegative and takes zero if and only if both $\theta$ and $\zeta$ are *correct* with respect to $\theta^*$, i.e., $\delta(\theta^* \| \theta, \zeta) = 0$ if and only if

$$p(\mathbf{y}|\mathbf{u}, \theta) = p(\mathbf{y}|\mathbf{u}, \theta^*), \qquad q(\mathbf{z}|\mathbf{y}, \mathbf{u}, \phi) = p(\mathbf{z}|\mathbf{y}, \mathbf{u}, \theta),$$
$$q(\mathbf{z}|\mathbf{u}, \psi) = p(\mathbf{z}|\mathbf{u}, \theta), \qquad\qquad f(\mathbf{u}; \xi) = \ln p(\mathbf{u}|\theta),$$

for all $\mathbf{y}$, $\mathbf{u}$ and $\mathbf{z}$. Note that the correctness of $(\theta, \zeta)$ is measured with respect to the *predictive form* of the true model $p(\mathbf{y}|\mathbf{u}, \theta^*)$, not the generative form $p(\mathbf{y}, \mathbf{u}|\theta^*)$, which is an essence of DIG. In other words, $\delta(\theta^* \| \theta, \zeta)$ measures the predictive discrepancy of $(\theta, \zeta)$ from $\theta^*$.

**Justification of CELBO maximization.**   Observe that

$$\mathbb{E}_{(\mathbf{y},\mathbf{u}) \sim \theta^*}[\hat{\mathcal{L}}_{\text{CELBO}(\mathbf{y}|\mathbf{u})}(\theta, \zeta)] = -h(\mathbf{y}|\mathbf{u}) - \delta(\theta^* \| \theta, \zeta),$$

where $h(\mathbf{y}|\mathbf{u}) := \mathbb{E}_{\theta^*}[-\ln p(\mathbf{y}|\mathbf{u}, \theta^*)]$ is the differential entropy of $\mathbf{y}$ given $\mathbf{u}$. Since $h(\mathbf{y}|\mathbf{u})$ is independent of $(\theta, \zeta)$, the CELBO maximization is justified as the discrepancy-function minimization,

$$\underset{(\theta,\zeta) \in \Omega}{\text{maximize}} \; \mathbb{E}_{(\mathbf{y},\mathbf{u}) \sim \theta^*}[\hat{\mathcal{L}}_{\text{CELBO}(\mathbf{y}|\mathbf{u})}(\theta, \zeta)] \quad \Leftrightarrow \quad \underset{(\theta,\zeta) \in \Omega}{\text{minimize}} \; \delta(\theta^* \| \theta, \zeta).$$

**Justification of CELBO-SP maximization as surrogate.**   Similarly, we have

$$\mathbb{E}_{(\mathbf{y},\mathbf{u}) \sim \theta^*}[\hat{\mathcal{L}}_{\text{CELBO-SP}(\mathbf{y}|\mathbf{u})}(\theta, \zeta)] = -h(\mathbf{y}|\mathbf{u}) - \delta_{\text{SP}}(\theta^* \| \theta, \zeta),$$

where $\delta_{\text{SP}}(\theta^* \| \theta, \zeta) := \delta(\theta^* \| T_{\text{SP}}(\theta, \zeta))$ is referred to as the *transformed* discrepancy function. Then, the CELBO-SP maximization is seen as the transformed-discrepancy-function minimization,

$$\underset{(\theta,\zeta) \in \Omega}{\text{maximize}} \; \mathbb{E}_{(\mathbf{y},\mathbf{u}) \sim \theta^*}[\hat{\mathcal{L}}_{\text{CELBO-SP}(\mathbf{y}|\mathbf{u})}(\theta, \zeta)] \quad \Leftrightarrow \quad \underset{(\theta,\zeta) \in \Omega}{\text{minimize}} \; \delta_{\text{SP}}(\theta^* \| \theta, \zeta).$$

Moreover, the transformed discrepancy function is consistent with the original discrepancy function in the following sense.

**Proposition 5.** *For all $\theta^*$, $\theta$ and $\zeta$,*

$$\delta(\theta^* \| \theta, \zeta) = 0 \quad \Leftrightarrow \quad \delta_{\text{SP}}(\theta^* \| \theta, \zeta) = 0. \tag{18}$$

*Proof.* It is shown as a corollary of Proposition 2. □

In other words, $\delta_{\text{SP}}(\theta^* \| \theta, \zeta)$ also measures the predictive discrepancy of $(\theta, \zeta)$ from $\theta^*$ in a different way. This justifies the CELBO-SP maximization as a surrogate of the CELBO maximization.

**Limitation of CELBO-SP maximization as surrogate.**   A property stronger than the consistency (Equation 18) is the *domination* of the discrepancy function. Here, we say $\delta_{\text{SP}}$ dominates $\delta$ if there exists $C < \infty$ such that

$$\delta(\theta^* \| \theta, \zeta) \leq C \delta_{\text{SP}}(\theta^* \| \theta, \zeta). \tag{19}$$

for all $\theta^*$ and $(\theta, \zeta) \in \Omega$. If the domination holds, then the CELBO-SP maximization implies (in expectation) the CELBO maximization up to a multiplicative constant.

Unfortunately, however, this is not the case in general. It is partly by design since the transformed discrepancy function $\delta_{\mathrm{SP}}(\theta^*\|\theta, \zeta)$ is derived as a result of suppressing the divergence of the density ratio $w_{\theta,\psi,\xi}(\mathbf{u}, \mathbf{z})$, which also causes the divergence of the original discrepancy function $\delta(\theta^*\|\theta, \zeta)$. The following proposition shows that there exists no such constant $C < \infty$ satisfying Equation 19.

**Proposition 6.** *There exists a triple $(\theta^*, \theta, \zeta)$ such that $\delta(\theta^*\|\theta, \zeta) = \infty$ and $\delta_{\mathrm{SP}}(\theta^*\|\theta, \zeta) < \infty$.*

*Proof.* It suffices to take $(\theta^*, \theta, \zeta)$ such that $w_{\theta,\psi,\xi}^{\alpha}(\mathbf{u}, \mathbf{z})$ is not integrable with the density $q(\mathbf{z}|\mathbf{u}, \psi)$, but $\{w_{\theta,\psi,\xi}(\mathbf{u}, \mathbf{z})G(w_{\theta,\psi,\xi}(\mathbf{u}, \mathbf{z}))\}^{\alpha}$ is integrable with the same density. For example, assume $\mathbf{z}$ takes a value in a Euclid space and take $p(\mathbf{z}|\mathbf{u}, \theta) \propto \exp(-\|\mathbf{z}\|)$ and $q(\mathbf{z}|\mathbf{u}, \psi) \propto \exp(-\|\mathbf{z}\|^2)$. $\quad\square$

