# OpenReview forum: "Variational Inference for Discriminative Learning with Generative Modeling of Feature Incompletion"
_ICLR.cc/2022/Conference — ICLR 2022 Oral_

### Official Review · Reviewer_WZsz · 2021-10-31

**Correctness:** 4
**Technical Novelty And Significance:** 3
**Empirical Novelty And Significance:** 3
**Recommendation:** 6
**Confidence:** 4

**Main Review:**

Strengths:

1. The idea of learning missing data using discriminative learning together with generative modeling is interesting. As mentioned in the paper, performing such kind of learning will resulting a loss function as a subtraction on two integrals with respect to the latent variables, which makes it harder to derive a lower bound compared to the traditional variational inference cases. To solve the term being subtracted, the authors found a upper bound that could be estimated in an unbiased way with Monte Carlo methods.

2. The exponential function in the first version of CELBO will potentially has a bigger variance when estimated using Monte Carlo. To solve this issue, the authors propose adding a regularization to the loss function while remain the optimal solution unchanged under the zero gap case. This also helps a lot in making training more stable, as shown in the experiments.

Weakness:

1. The topic of this paper is to focus on missing data. However, this paper does not put enough efforts on learning various cases of data missing patterns. As in the experiments, the authors only test the case where the data are missing completely at random (MCAR), which may not be the most common case in reality. MNAR case might be a more interesting situation to study with. The proposed method is mostly focusing on solving the variational upper bound, while overlooks modeling the missing patterns. Suggest the authors could add some experiments with MNAR data. Also it would be better if the authors could add some modeling part on the missing pattern into the loss. For example, let the mask $m$ to depends on $(x, y, z)$. This will make the proposed model more useful in practice.

2. From Proposition 2, we know that the surrogate parameterization can make the optimal solution of CELBO remains under the zero gap case. However, it is nearly impossible to reach the zero gap case in reality since it is unlikely to select variational distributions (i.e. $q(\cdot)$'s) to perfectly estimate the model posterior distributions. What about the "sub-optimal" cases? Is the CELBO-SP optimal solution close to the CELBO optimal solution in a small gap (but not zero gap) case? I understand this might not be easy, but it will be better if the authors could add some theoretical analysis on this.

3. The performance metrics in Table 1 is not showing the proposed methods can outperform the baselines with big gaps, meaning that the proposed methods is not much better compared to previous approaches empirically. However, I think there are many ways that the authors could try to improve the performances. For example, the authors could try a different divergence function, a better way to add the regularization, etc.

**Summary Of The Paper:**

This paper proposes a new method for learning with missing data. Compared with previous approaches, the authors choose to perform discriminative learning with generative modeling so as to borrow the benefits from these two types of methods. To optimize with the underlying intractable loss function, the authors start from the traditional variational lower bound ELBO and one upper bound CUBO from a previous work (the $\chi$-divergence lower bound [1]) and derives a lower bound for the original loss function. To solve the issue with the estimation bias as well as the potential huge variance, the authors change the divergence function in CUBO as well as add the surrogate parameterization so that the Monte Carlo estimation of the loss can be unbiased and (potentially) with smaller variances. Experiment results show the proposed methods run stably and perform comparably or better compared to baseline methods.

References:

[1] Adji Bousso Dieng, Dustin Tran, Rajesh Ranganath, John Paisley, and David Blei. Variational inference via χ upper bound minimization. In Advances in Neural Information Processing Systems, pp. 2732–2741. 2017.

**Summary Of The Review:**

The author propose an interesting discriminative learning approach with generative modeling to solve the missing data modeling problem, by extending the traditional variational lower bound (ELBO), with a novel and stable upper bound that can be estimated without bias with Monte Carlo estimation. It is better if the author can study more on the missing data pattern (both empirically and theoretically) and the optimal solution preservation (under sub-optimal cases). Also, the empirical performance still has some space to improve.

---

> ### Author Response · Authors · 2021-11-16
> **Thank you**
>
> Thank you for the helpful comments and suggestions.
> See below for the answers to your questions and comments
> (before proceeding to the updated manuscript, note that we have simplified a part of the notation, denoting the pair of observable features $(\tilde{x}, m)$ with $u$).
>
>
> > The topic of this paper is to focus on missing data. However, this paper does not put enough efforts on learning various cases of data missing patterns. As in the experiments, the authors only test the case where the data are missing completely at random (MCAR), which may not be the most common case in reality. MNAR case might be a more interesting situation to study with. The proposed method is mostly focusing on solving the variational upper bound, while overlooks modeling the missing patterns. Suggest the authors could add some experiments with MNAR data.
>
> We are willing to conduct additional experiments with MNAR datasets.
> Please wait until we figure out if we can carry it out within the discussion period.
>
>
> > Also it would be better if the authors could add some modeling part on the missing pattern into the loss. For example, let the mask $m$ to depends on $(x, y, z)$. This will make the proposed model more useful in practice.
>
> We have generalized the notation to allow such flexible modeling of missing data.
> We have also added some explicit examples of missing data modeling.
> Please refer to Section 2.2.
>
>
> > From Proposition 2, we know that the surrogate parameterization can make the optimal solution of CELBO remains under the zero gap case. However, it is nearly impossible to reach the zero gap case in reality since it is unlikely to select variational distributions (i.e. $q(\cdot)$'s) to perfectly estimate the model posterior distributions. What about the "sub-optimal" cases? Is the CELBO-SP optimal solution close to the CELBO optimal solution in a small gap (but not zero gap) case? I understand this might not be easy, but it will be better if the authors could add some theoretical analysis on this.
>
> Right, the condition of the exact tightness is not likely to be achieved in practice with
> complex parameter spaces such as those induced by neural nets.
> However, we still believe it is sufficient to give some justification on the use of CELBO-SP as a surrogate of CELBO.
> Please refer to Section H, which
> we have added for the clarification and additional discussion on this point.

---

> > ### Author Response · Authors · 2021-11-22
> > **On additional experiment**
> >
> > We have decided to leave the MNAR experiment as future work
> > considering the necessity and the time constraint.
> >
> > Note that MCAR is already suitable for demonstrating
> > an advantage of vDIG since it can be captured only with the generative approach and the DIG approach,
> > not with the discriminative approach.
> > On the other hand, MNAR in general can be captured with all of these approaches
> > and we need to make careful design choices to specify subclasses of MNAR meaningful for the comparison.
> >
> > Thus, it is an interesting direction for future study
> > to identify specific types of the feature corruption process relevant to some applications
> > and see if vDIG can be useful to solve the associated prediction problems.

---

### Official Review · Reviewer_vurp · 2021-11-01

**Correctness:** 4
**Technical Novelty And Significance:** 3
**Empirical Novelty And Significance:** 3
**Recommendation:** 6
**Confidence:** 3

**Main Review:**

**Strengths**
The paper is overall clearly written. The issue it tries to solve, discriminative tasks with missing input features, has great impact for a wide range of practical machine learning problems in real life. Technically, the paper has quite some novelty including the creation of a rigorous lower bound to the true objective using recent advances in the variational inference area, and designed an effective surrogate parameterization to stabilize the optimization.

**Weaknesses and Questions**
1. Sec. 3.1: More detailed explanation of the exponential divergence would be beneficial. Is there a reference for it? What role does $f(\boldsymbol{u}; \xi)$ play? If it can be any real-valued function, why was it chosen to be a Gaussian pdf, as shown in the appendix?
2. Sec. 3.2: Which standard automatic differentiation library was used? The submission mentioned both the reparameterization trick and the REINFORCE trick - which one was actually used in the experiments?
3. Sec. 3.3: I don't quite understand how this part works. All I can see that in Eq(17) the problematic ratio term is multiplied by $G$, which is always smaller than 1 and non-increasing based on Figure 3. But why $G$ was defined in that math format? What does $\vee$ mean? Why does $G$ represent the ratio before and after the transform (equation between proposition 2 and 3)?
4. Sec. 4.1: Missing value processes (MCAR and MNAR). What do they mean? Are they different ways to decide what values are missing in the feature, and thus leading to different versions of a dataset? If so, shouldn't we also add CVAE*, Simple* and MICE*? Could you give more explanation for the last sentence of Section 4 (saying DVAE* is more robust than DVAE)?
5. Size of the test datasets. Based on Table 3, the datasets are all quite small, ranging from 353 to 10k data points. And we further split these points into training and test, which makes the training sets even smaller. In the appendix it's said the minibatch size is 521 -- what if the entire training set is smaller than 512? How long did the algorithm take to run on YearPred? Is the algorithm able to be easily extended to larger datasets?
6. How does DIG works compared with other more recent imputation baselines such as MIWAE (Mattei & Frellsen, 2019) and GAIN (Yoon et al., 2018)?

**Summary Of The Paper:**

The authors propose a new method, DIG, for discriminative tasks with missing input features. It uses latent variable models to marginalize out the label and the missing part of the features given the latent variable, in order to compute the objective, the conditional log likelihood of the label given the corrupted features. As this objective is intractable, the paper builds a conditional evidence lower bound (CELBO) that can be unbiasedly approximated using Monte Carlo samples. CELBO consists of the regular ELBO as the lower bound for the log joint probability of the label and the observed features, and an evidence upper bound (EUBO) that bounds the log marginal probability of the observed features. The derivation of EUBO involves the alpha-renyi divergence and the exponential divergence. The stochastic CELBO contains a density ratio that can lead to large variance in the stochastic gradients during optimization, so the authors propose a surrogate parameterization to bound the gradient norm. Experiments on real datasets justify the effectiveness of the variational approximations to stabilize the optimization. When compared with VAE, CVAE, and MICE, the DIG algorithm shows better or comparable predictive performance and robustness against feature corruption.

**Summary Of The Review:**

Given the strengths of the paper listed above, I would recommend acceptance for this paper, if the authors can figure out a clear feedback for the questions I summarized when reading the paper.

---

> ### Author Response · Authors · 2021-11-16
> **Thank you**
>
> Thank you for the helpful comments and suggestions.
> See below for the answers to your questions and comments
> (before proceeding to the updated manuscript, note that we have simplified a part of the notation, denoting the pair of observable features $(\tilde{x}, m)$ with $u$).
>
>
> > Sec. 3.1: More detailed explanation of the exponential divergence would be beneficial. Is there a reference for it? What role does $f(u;\xi)$ play? If it can be any real-valued function, why was it chosen to be a Gaussian pdf, as shown in the appendix?
>
> The name 'the exponential divergence' does not refer to specific existing divergences.
> It is just a function used to derive EUBO.
> Given your feedback, we think giving it a name is confusing and decided to drop it.
>
> The role of $\Psi_\alpha$ can be understood as giving linear upper bound approximation of likelihood.
> In this context, the role of $f(u;\xi)$ is then understood as a parameter of the tangent point,
> $x_0=e^{\alpha f(u;\xi)}$.
>
> As for taking $f(u;\xi)$ as Gaussian pdf in the appendix, it is a typo.
> It should be a linear layer with a single output, not a Gaussian layer.
> Sorry for making confusion. We have fixed it.
>
>
> > Sec. 3.2: Which standard automatic differentiation library was used? The submission mentioned both the reparameterization trick and the REINFORCE trick - which one was actually used in the experiments?
>
> We used the reparametrization trick (updated Section B, First-order optimization).
>
>
> > Sec. 3.3: I don't quite understand how this part works. ... But why $G$ was defined in that math format? What does $\vee$ mean? Why does $G$ represent the ratio before and after the transform (equation between proposition 2 and 3)?
>
> The specific form of $G$ is chosen to obtain the desired properties (i.e., Proposition 2 and 3).
> We design it to be smooth and cause minimal, yet sufficient change in the density ratio $w$ to derive Proposition~3.
> $a\vee b$ denotes the pairwise maximum of $a$ and $b$ (sorry for the lack of the definition, we have fixed it).
> $G$ representing the ratio before and after the transform is shown by plugging the transformed parameters $(\theta',\psi,\xi')$ into the definition of the density ratio $w_{\theta,\psi,\xi}(u,z)$.
>
> > Sec. 4.1: Missing value processes (MCAR and MNAR). What do they mean? Are they different ways to decide what values are missing in the feature, and thus leading to different versions of a dataset? If so, shouldn't we also add CVAE\*, Simple\* and MICE\*? Could you give more explanation for the last sentence of Section 4 (saying DVAE\* is more robust than DVAE)?
>
> Sorry, we mistakenly dropped the citation there (added in the new manuscript: see Section 2.2).
> They are standard types of the assumptions on the missing value mechanism given by Rubin 1976.
> In short, MNAR is no assumption while MCAR is a strong assumption that missing value is independent of $x,y$.
> Since MCAR is an assumption on the process of feature corruption,
> it is impossible (at least in a trivial way) to incorporate it in discriminative methods such as CVAE and Simple (note that this is the major motivation behind the DIG method).
> On the other hand, MICE is derived under another assumption, namely MAR,
> so there is no MCAR and MNAR variants.
>
> The last sentence of Section 4 saying the DVAE\* is more robust than DVAE
> is reflecting the fact that MCAR is a stronger and correct assumption (in this setting) than MNAR.
>
>
> > Size of the test datasets. ... In the appendix it's said the minibatch size is 521 -- what if the entire training set is smaller than 512? How long did the algorithm take to run on YearPred? Is the algorithm able to be easily extended to larger datasets?
>
> If the data size is smaller than 512, then the minibatch size is reduced to the data size.
> The training time of DVAE for YearPred was 35.9 seconds.
>
>
> > How does DIG works compared with other more recent imputation baselines such as MIWAE (Mattei & Frellsen, 2019) and GAIN (Yoon et al., 2018)?
>
> Thanks for the pointers to the reference.
>
> The contribution of variational algorithms for likelihood-based generative models, including MIWAE, is orthogonal to ours.
> In particular, since MIWAE is an algorithm that gives tighter lower bound for missing-data likelihoods,
> it can be combined with vDIG at least to improve the first variational posterior $q(z|y,\tilde{x},m,\phi)$.
> The applicability to the second posterior $q(z|\tilde{x},m,\psi)$ is more intricate since
> $\psi$ is a parameter for the upper bound approximation, which is out of MIWAE's (and most of other VI methods') scope.
>
> On the other hand, the comparison/combination with GAIN is less straightforward
> since GAIN's objective function is not the log likelihood, but the adversarial error.
> We would like to left it as future work.

---

> > ### Comment · Reviewer_vurp · 2021-11-21
> > **Questions on the assumptions in the generative process and the dataset scales**
> >
> > Thank you for clarifying all my questions.
> >
> > The modification about adding the generative process for MCAR, MNAR and MAR is very helpful. I'm wondering why the conditional distribution of label $y$ is always irrelevant of the observed features $x$ in all the three cases. Is this part of the constrains you set in the assumption of the generative process?
> >
> > From the authors' response, it only took less than a minute to train on the largest dataset (YearPred) used in the submission. Why the algorithm was not tested on any larger datasets? The datasets in table 3 are all at very small scale.

---

> > > ### Author Response · Authors · 2021-11-22
> > > **On generative process and dataset scales**
> > >
> > > Thank you for additional feedbacks and questions.
> > >
> > > > The modification about adding the generative process for MCAR, MNAR and MAR is very helpful. I'm wondering why the conditional distribution of label $y$ is always irrelevant of the observed features $x$ in all the three cases. Is this part of the constrains you set in the assumption of the generative process?
> > >
> > > In our formulation, the dependence between $x$ and $y$ is captured through $z$. Note that Equation 5 and 6 are *joint* distribution of ($x$, $y$, $z$). The distribution of $y$ is relavant to the feature $x$ after $z$ is marginalized. This kind of independence among varialbes in the $z$-*joint* density is common in the VAE literature, as in the original paper (Kingma & Welling, 2013).
> > >
> > > > From the authors' response, it only took less than a minute to train on the largest dataset (YearPred) used in the submission. Why the algorithm was not tested on any larger datasets? The datasets in table 3 are all at very small scale.
> > >
> > > The reason is that
> > > the scalability of vDIG is not the primary subject of our experiment.
> > > In particular, the scalability of DVAE is inferable from those of VAE and CVAE
> > > since they are all SGD-based and the numbers of the parameters are not that different owing to DVAE's parameter sharing among $\phi$, $\psi$ and $\xi$.
> > > See below for more information.
> > >
> > > ### Computation time comparison
> > >
> > > To demonstrate how the scalability of DVAE is inferable,
> > > we report the training time (only gradient computation and parameter update are accounted, the saving/evaluation times are ignored)
> > > of each method for YearPred.
> > > It is seen that DVAE costs around 2.25x CVAE or 1.2x VAE.
> > > The reason of DVAE being almost twice as expensive as CVAE is, supposedly, because
> > > there are two encoding-decoding computation paths (i.e., one with $z_\phi$ and one with $z_\psi$).
> > > It is slightly more expensive than twice (2.25x > 2x), which is attributed to the unsharable parameters of $\psi$ and $\xi$.
> > > On the other hand, the reason of VAE being almost twice as expensive as CVAE is also because
> > > there are two paths corresponding to the doubled minibatches for learning the imputation of $y$ (see Section 4.1).
> > > Note that the training time of MICE is not reported
> > > since we did not measure the time of imputation with MICE (we only measured the time for the gradient computation and parameter updates *after* MICE, which is the same as Simple).
> > >
> > > |       |     Time [sec] |
> > > |:------|---------:|
> > > | DVAE  | 35.9171  |
> > > | DVAE* | 35.1979  |
> > > | CVAE  | 16.3529  |
> > > | VAE   | 30.6778  |
> > > | VAE*  | 29.8359  |
> > > | Simple |  7.51111 |
> > >
> > > ### Total execution time of the current setup
> > >
> > > The current experimental setup already costs ~10 hours to execute in total.
> > > There are multiple factors that blow up the total execution time.
> > > First, the bottleneck of our experiment is in fact not the training time,
> > > but the time to save/evaluate the model for each 20 checkpoints per single run (the time we have just reported, 39.5 sec, does not include this).
> > > Note that placing 20 checkpoints are rather overdoing for a single algorithm,
> > > but necessary for fair comparison among different algorithms with different optimal early-stopping timings.
> > > Moreover, if we adopt larger datasets, then probably we need more checkpoints.
> > > Another factor is the training time of MICE, whose scalability is the worst supposedly because of its CPU-based implementation.
> > > Finally, recall that the total time of the experiment is (roughly) multiplied with
> > > the number of configurations, namely, (7 methods) x (6 datasets) x (5 random seeds) x (2 missing-value ratios) = 420 configurations.

---

### Official Review · Reviewer_7Sj3 · 2021-11-01

**Correctness:** 4
**Technical Novelty And Significance:** 4
**Empirical Novelty And Significance:** 3
**Recommendation:** 8
**Confidence:** 4

**Main Review:**

The ab-initio generative model class proposed by the authors for handling predictions with missing features is convincing. It has the advantage that the involved distributions are simple (factorising), however at the price of introducing latent variables. To learn it discriminatively requires to maximise a difference of concave functions. The first term is lower bounded by ELBO as in VAEs. The second term requires a tractable upper bound. The authors develop a  novel upper bound (starting from alpha-Rényi divergence) that admits a stochastic gradient estimator.  They further introduce a data dependent surrogate reparametrisation in order to achieve an estimator with low variance. The technical part of the paper is concisely written and correct.

The authors prove that the transformation used for the reparametrisation preserves the effective parameter subset, i.e. the subset of parameter combinations for which the overall bound is tight. This is indeed a desirable property, but is in my view not sufficient. The reason is, that this effective subset can be very small and cover a subset of simple models only. Moreover, there is no guarantee that the respective gap will become small during learning.

The experimental section first analyses the learning properties of the method in an ablation study. The authors then show competitiveness of their method by comparing it with existing approaches on a subset of tasks taken from the  UCI Machine Learning Repository. The description of the experiments is clear and reproducible. The experiments are however not fully convincing w.r.t. the scalability of the approach. All networks used for the model and bound construction have only one fully connected hidden layer. This seems to be sufficient for the considered tasks from the  UCI repository. However, this would be not sufficient e.g. for image classification tasks where the involved networks are usually deep CNNs.

Further comments:
- You mention earlier works  (Ghahramani & Jordan, 1994; Smola et al., 2005), noting that their applicability is restricted to exponential families. Please explain whether these approaches are / are not applicable for the model class analysed in your work. As I understand it, the models p(y,x,z|m) considered by you are exponential families, but of course after marginalising over z, the resulting mixture model p(y,x|m) is not any more. It remains however unclear to me, whether a DCA (difference of convex functions algorithm) for learning p(y|x,z,m) can be somehow generalised for learning p(y|x,m).

- I would suggest to drop the data instance superscript earlier in the text, e.g. starting from subsection 2.2. the latest. This would in my view improve readability and reduce clutter.

**Summary Of The Paper:**

The paper considers the problem of prediction with missing (incomplete) features. The authors propose a class of generative models that includes missingness of features, and develop a discriminative learning algorithm that maximises the conditional (posterior) log-likelihood of the training data approximately. Experiments show that the method is competitive compared with existing approaches, and in particular with approaches based on VAEs.

**Summary Of The Review:**

The conceptual part, i.e. the model and the proposed learning approach are in my view concise and sufficiently novel. This outweighs the missing scalability analysis in the experimental part. I would however expect the authors to clearly address the raised conceptual questions.

---

> ### Author Response · Authors · 2021-11-16
> **Thank you**
>
> Thank you for the helpful comments and suggestions.
> See below for the answers to your questions and comments
> (before proceeding to the updated manuscript, note that we have simplified a part of the notation, denoting the pair of observable features $(\tilde{x}, m)$ with $u$).
>
> > The authors prove that the transformation used for the reparametrisation preserves the effective parameter subset, i.e. the subset of parameter combinations for which the overall bound is tight. This is indeed a desirable property, but is in my view not sufficient. The reason is, that this effective subset can be very small and cover a subset of simple models only. Moreover, there is no guarantee that the respective gap will become small during learning.
>
> Right, the condition of the exact tightness is not likely to be achieved in practice with
> complex parameter spaces such as those induced by neural nets.
> However, we still believe it is sufficient to give some justification on the use of CELBO-SP as a surrogate of CELBO.
> Please refer to Section H, which
> we have added for the clarification and additional discussion on this point.
>
>
> > You mention earlier works (Ghahramani & Jordan, 1994; Smola et al., 2005), noting that their applicability is restricted to exponential families. Please explain whether these approaches are / are not applicable for the model class analysed in your work. As I understand it, the models p(y,x,z|m) considered by you are exponential families, but of course after marginalising over z, the resulting mixture model p(y,x|m) is not any more. It remains however unclear to me, whether a DCA (difference of convex functions algorithm) for learning p(y|x,z,m) can be somehow generalised for learning p(y|x,m).
>
> The models $p(y,x,z|m,\theta)$ can be more general than exponential families.
> Specifically, in the example of VAE, $y$ and $x$ depends on an NN-based transformation of $z$ with weight parameters $\theta$,
> which implies it is not exponential family in general.
> In fact, both Ghahramani & Jordan, 1994 and Smola et al., 2005 solve the optimization of the mixture of exponential families
> either with the EM or DCA algorithm.
>
>
> > I would suggest to drop the data instance superscript earlier in the text, e.g. starting from subsection 2.2. the latest. This would in my view improve readability and reduce clutter.
>
> Thanks for the suggestion. We have updated the manuscript accordingly.

---

> > ### Comment · Reviewer_7Sj3 · 2021-11-19
> > **Further response**
> >
> > > Right, the condition of the exact tightness is not likely to be achieved in practice with complex parameter spaces such as those induced by neural nets. However, we still believe it is sufficient to give some justification on the use of CELBO-SP as a surrogate of CELBO. Please refer to Section H, which we have added for the clarification and additional discussion on this point.
> >
> > Thank you for clarifying this and adding Section H in the supplementary material.
> >
> > > The models $p(y,x,z|m)$ can be more general than exponential families. Specifically, in the example of VAE, $y$ and $x$ depends on an NN-based transformation of $z$ with weight parameters , which implies it is not exponential family in general. In fact, both Ghahramani & Jordan, 1994 and Smola et al., 2005 solve the optimization of the mixture of exponential families either with the EM or DCA algorithm.
> >
> > It seems to me that standard VAEs assume both the decoder distribution $p(x|z;\theta)$ and the encoder distribution $q(z|x;\phi)$ to be exponential families. If we denote e.g. the decoder network by $d(z,\theta)$, we have
> > $$ p(x|z;\theta) = \exp[<\psi(x),d(z,\theta)> - A(x,\theta)]$$
> > where $\psi(x)$ denotes the sufficient statistics of the family. Thus, the decoder output represents the natural parameter vector as a function of its parameters $\theta$. I understand that this "re-parametrisation" prevents a direct application of the methods from  Ghahramani & Jordan, 1994 and Smola et al., 2005. However, I would not exclude that something can be done here.
> >
> > Thank you for your comments and clarifications. I will keep my score (accept).

---

### Official Review · Reviewer_unNG · 2021-11-03

**Correctness:** 4
**Technical Novelty And Significance:** 3
**Empirical Novelty And Significance:** 1
**Recommendation:** 5
**Confidence:** 3

**Main Review:**

(+) pros / (-) cons
-------------------

(+) Predictive modelling over inputs with missing features is an important problem arising often in many application domains. This paper contributes this somewhat underexplored field.

(-) The rather low documented performance benefits over simpler baselines do not justify the use of the complex model (combining 5? networks) proposed here as opposed to simpler VAE or CVAE variants.

(+) The method and the various bounds introduced are mathematically intriguing, well motivated and potentially useful in follow-up research, however, ...
(-) the paper is difficult to  follow and at places the reader is left guessing what the authors meant. This should be improved. Concretely
1. last para of section 2 - the optimization of negative L "is relatively difficult". Why? What makes it difficult?
2. last para of section 2 - "... have been no equivalent of VI ... " What about CUBO and its variants pick up from in your work? These have some specific flaws for which they do not qualify here?
3. before equation 3 - "exponential divergence". You mean the Bregman exponential divergence? A citation to help the reader?
4. $p(u, z | \theta)$ in equations (11) and (12) seem to use the same parameters $\theta$ though for (11) $u = (y, \tilde{x})$ and for (12) it is $u = \tilde{x}$. Is this in practice the same network with two outputs?
5. But then in equation (15) these use different $z_{\theta}$ and $z_{\psi}$ samples. How is this designed and trained in practice?
6. page 5 - clarify notation for and explain the gain function; what is the intuition / purpose for it?
7. Def 1 - effective parameters are those with gradient zero. ".. i.e. the set of parameters inducing tight variational approximation." How zero gradient achieves this in a complex non-convex problem, i.e. can't this be a local non-tight extremum?
8. page 7 DVAE/ DVAE* - you say these are MNAR and MCAR model variants as in Collier at al. 2020. Can you clarify how these translate into your rather more complex model formulation and what specifically changes in the loss (especially the EUBO part)?

Further questions for clarification/discussion
-----------------------------------------------

1) Why do you condition $y$ and $x$ on $m$ in equation (1). These are the complete $x$ data so they should not depend on the masking so that $p(y, x | m) = p(y, x)$. Or is this not true? Or is it the $y$ that depends on $m$? Or is it rather the $m$ which depends on $x$? (As in some values being more likely to be masked?)

2) You introduce two latent variable models in equations (7) and (8). My understanding is that the latent $z$ is shared as (8) is just a marginalization of (7) over $y$. You then formulate to approximate posteriors $q(z| y, \tilde{x})$ and $q(z| \tilde{x})$ the first learned through ELBO maximization, the 2nd through EUBO minimization. There is currently no link between the two (approximate) posteriors. Would it make sense to somehow link them? (Sorry, I don't know how and may not be possible, or not easily.)

3) You used the Bayes rule to decompose the predictive conditional log likelihood into two terms in equation (4) of which one you are bounding from the bottom (ELBO) and the oher from top (EUBO). What is the effect on the predictive conditional $p(y | x)$? Is it somehow sandwiched or not really due to splitting and modelling the two non-conditional log likelihoods separately?


Minor text problems / typos
---------------------------

1. This first proposition in in page 6 is numbered 2 (not 1) - confusing

**Summary Of The Paper:**

The paper addresses the problem of predictive modelling with missing input features. The authors formulate the problem as a latent variable model, and in addition to the standard variational lower bound (ELBO) propose to use a variational upper bound based on CUBO (Dieng at al 2017) modified by an exponential divergence to solve the MC estimation of CUBO. They further propose surrogate parametrization to reduce the variance in the gradients. The experimental evaluation over standard regression UCI datasets with randomly dropped features shows marginal improvements over existing baselines.

**Summary Of The Review:**

The paper contributes to practically very important yet relatively little explored area of research - that of predictive modelling with missing data. The proposed method is rather complex, composed of multiple steps adding onto each other to solve a problem arising in the previous steps. These are all well motivated, however, overall the current presentation of the method is difficult to follow and should be improved to help the reader (see main review). Moreover, the documented performance benefits seem to be rather little to justify the use of such a complex method over simpler baseline. These two (lack of clarity, low performance given the complexity of method) are for me the reasons not to consider the paper for this conference.

---

> ### Author Response · Authors · 2021-11-16
> **Thank you**
>
> Thank you for the helpful comments and suggestions.
> We have updated the manuscript to make it explicit on the intentions and intuitions.
> See below for the answers to your questions and comments
> (before proceeding to the updated manuscript, note that we have simplified a part of the notation, denoting the pair of observable features $(\tilde{x}, m)$ with $u$).
>
>
> > last para of section 2 - the optimization of negative L "is relatively difficult". Why? What makes it difficult?
>
> There is a typo. The 'and' just after the phrase must be 'since'.
> Thus, the reason is because there have been no upper bound approximation of the evidence comparable
> to the VI for lower bounds in terms of scalability and flexibility.
>
>
> > last para of section 2 - "... have been no equivalent of VI ... " What about CUBO and its variants pick up from in your work? These have some specific flaws for which they do not qualify here?
>
> We discussed it in Section 5. We have placed a pointer there.
>
>
> > before equation 3 - "exponential divergence". You mean the Bregman exponential divergence? A citation to help the reader?
>
> The name 'the exponential divergence' does not refer to specific existing divergences.
> It is just a function used to derive EUBO.
> Given your feedback, we think giving it a name is confusing and decided to drop it.
>
>
> > $p(u,z|\theta)$ in equations (11) and (12) seem to use the same parameters $\theta$ [...]. Is this in practice the same network with two outputs?
>
> It could be. In the experiment, we employed the network structure you mentioned (see Appendix D).
> However, the proposed algorithm is designed without such structural assumptions.
> For example, (11) and (12) (in the updated manuscript, (10) and (11)) can be implemented with different networks without any parameter sharing.
> In this case, $\theta$ represents the union of the parameters of both networks.
>
>
> > But then in equation (15) these use different $z_\phi$ and $z_\psi$ samples. How is this designed and trained in practice?
>
> It is trained by any stochastic gradient-based optimization algorithms.
> We believe there is no ambiguity in the procedure with/without the aforementioned parameter sharing.
>
>
> > page 5 - clarify notation for and explain the gain function; what is the intuition / purpose for it?
>
> It is detailed in the next paragraph. We have added a pointer there.
> The intuition is to suppress the divergence of the problematic density ratio.
>
>
> > Def 1 - effective parameters are those with gradient zero. ".. i.e. the set of parameters inducing tight variational approximation." How zero gradient achieves this in a complex non-convex problem, i.e. can't this be a local non-tight extremum?
>
> It is not the gradient.
> It denotes the gap of the variational approximation (defined along with CELBO, see the paragraph just after equation (11)).
>
>
> > page 7 DVAE/ DVAE\* - you say these are MNAR and MCAR model variants as in Collier at al. 2020. Can you clarify how these translate into your rather more complex model formulation and what specifically changes in the loss (especially the EUBO part)?
>
> We have added explanation on this point in Section 2.2. (and placed pointers in Secion 4)
> The difference between the MNAR and MCAR variants is the same between VAE-based and DVAE-based formulations,
> since both VAE and DVAE has the same decoder structure (the difference is in encoders)
> and the missing-value processes are modeled with the decoder.
>
>
> > Why do you condition $y$ and $x$ on $m$ in equation (1). These are the complete $x$ data so they should not depend on the masking so that $p(y,x|m)=p(y,x)$. Or is this not true? Or is it the $y$ that depends on $m$? Or is it rather the $m$ which depends on $x$? (As in some values being more likely to be masked?)
>
> In the general case of MNAR, both $x$ and $y$ can depend on $m$.
> We have added explanation on this point in Section 2.2.
>
>
> > You introduce two latent variable models in equations (7) and (8). ... There is currently no link between the two (approximate) posteriors. Would it make sense to somehow link them? (Sorry, I don't know how and may not be possible, or not easily.)
>
> Your suggestion is a possible extension of our formulation.
> Note, however, that these posterior approximations are already liked (edit: typo - *linked*) together indirectly
> as these are designed to approximate different aspects of the same generative model.
>
> > What is the effect on the predictive conditional $p(y|x)$? Is it somehow sandwiched or not really due to splitting and modelling the two non-conditional log likelihoods separately?
>
> There is no sandwich effect:
> The effect is that the predictive conditional log likelihood is bounded from below.

---

### Decision · Program_Chairs · 2022-01-20

**Decision:**

Accept (Oral)

**Comment:**

While generative model can be used to input data, this work propose to a novel discriminative learning approach to optimize this data imputation phase by deriving a discriminative version of the traditional variational lower bound (ELBO). The resulting bound can be estimated without bias with Monte Carlo estimation leads to a practical approach, leading to encouraging experimental performances.

The reviewers recognised the novelty and suggest that this approach, given its novelty and wide applicability, could be considered for an oral presentation.